# Antimicrobial Resistance in the Global Health Network: Known Unknowns and Challenges for Efficient Responses in the 21st Century

**DOI:** 10.3390/microorganisms11041050

**Published:** 2023-04-17

**Authors:** Teresa M. Coque, Rafael Cantón, Ana Elena Pérez-Cobas, Miguel D. Fernández-de-Bobadilla, Fernando Baquero

**Affiliations:** 1Servicio de Microbiología, Hospital Universitario Ramón y Cajal, Instituto Ramón y Cajal de Investigación Sanitaria (IRYCIS), 28034 Madrid, Spain; 2CIBER en Enfermedades Infecciosas (CIBERINFEC), Instituto de Salud Carlos III, 28029 Madrid, Spain; 3CIBER en Epidemiología y Salud Pública (CIBERESP), Instituto de Salud Carlos III, 28029 Madrid, Spain

**Keywords:** antimicrobial resistance, disease ecology, syndemic, antimicrobial stewardship, lateral public health, social health, global health, one health, ecosystem, metasystem

## Abstract

Antimicrobial resistance (AMR) is one of the Global Health challenges of the 21st century. The inclusion of AMR on the global map parallels the scientific, technological, and organizational progress of the healthcare system and the socioeconomic changes of the last 100 years. Available knowledge about AMR has mostly come from large healthcare institutions in high-income countries and is scattered in studies across various fields, focused on patient safety (infectious diseases), transmission pathways and pathogen reservoirs (molecular epidemiology), the extent of the problem at a population level (public health), their management and cost (health economics), cultural issues (community psychology), and events associated with historical periods (history of science). However, there is little dialogue between the aspects that facilitate the development, spread, and evolution of AMR and various stakeholders (patients, clinicians, public health professionals, scientists, economic sectors, and funding agencies). This study consists of four complementary sections. The first reviews the socioeconomic factors that have contributed to building the current Global Healthcare system, the scientific framework in which AMR has traditionally been approached in such a system, and the novel scientific and organizational challenges of approaching AMR in the fourth globalization scenario. The second discusses the need to reframe AMR in the current public health and global health contexts. Given that the implementation of policies and guidelines are greatly influenced by AMR information from surveillance systems, in the third section, we review the *unit of analysis* (“the what” and “the who”) and the indicators (the “operational units of surveillance”) used in AMR and discuss the factors that affect the validity, reliability, and comparability of the information to be applied in various healthcare (primary, secondary, and tertiary), demographic, and economic contexts (local, regional, global, and inter-sectorial levels). Finally, we discuss the disparities and similarities between distinct stakeholders’ objectives and the gaps and challenges of combatting AMR at various levels. In summary, this is a comprehensive but not exhaustive revision of the *known unknowns* about how to analyze the heterogeneities of hosts, microbes, and hospital patches, the role of surrounding ecosystems, and the challenges they represent for surveillance, antimicrobial stewardship, and infection control programs, which are the traditional cornerstones for controlling AMR in human health.

## 1. Antibiotic Resistance in the Global Healthcare Network: A Multifaceted Problem Involving Many Stakeholders of Disparate Sectors

*Antibiotic resistance* (AMR, see Appendix B for definitions of words in blue-italics) is recognized as one of the major Global Health challenges of the 21st century [1]. The inclusion of AMR in the global health map parallels the scientific, technological, and organizational progresses of the healthcare system and reflects the colossal socioeconomic changes of the last 100 years. Studies originating in various fields have examined particular aspects of the AMR problem, focused on patient safety (infectious diseases and infection control) [2], transmission pathways, and pathogen reservoirs (molecular epidemiology) [2,3,4], the extent of the problem at a population level (public health) [5,6,7], their management and cost (health economics) [8,9,10,11], behavioral aspects related to infection prevention, management, antibiotic prescription and antibiotic use (social sciences) [12,13], antimicrobial pollution (ecotoxicology and biodiversity) [14], and episodes associated with historical events (history of science) [15,16,17,18]. However, we lack an essential dialogue between all these aspects that have an important role in the development, transmission, and evolution of AMR. This work discusses the challenges of AMR in the global healthcare system, considering the heterogeneities of its (*ecosystem*) *structure*, the instruments employed to detect and monitor AMR, and the disparities and similarities of the objectives of distinct stakeholders. It also describes the changes in the theoretical framework and the effects of knowledge fragmentation in interpreting the AMR information throughout history.

### 1.1. The Modern Healthcare System: A Story of “Social Construction” with an Impact on AMR

Between the end of the 19th century and the early 20th century, medical treatment moved from providing care in particular houses, philanthropic organizations, religious charities (including primitive hospitals and quarantine stations), and ad hoc foundations for particular diseases to care centered in advanced hospitals [19]. This change was driven by industrialization and rapid urbanization which dramatically increased the numbers of poor, wounded, and ill people who had moved from the countryside to the cities and transformed charitable actions into a public health issue. The need for *therapeutic solutions* to counteract major epidemics of infectious diseases led to the birth of the “chemotherapy era” with the introduction of arsphenamine (Salvarsan) in 1910, sulfonamides in the 1930s, and natural antibiotics in the 1940s [15,16,18]. Advances in the pharmaceutical sector during these decades (such as the adoption of production models from the food industry, company alliances, and strong licensing restrictions by large Western companies) would influence the further development, production, and consumption of antibiotics in Western societies [20,21,22,23]. The birth and colossal development of health industries (hospital infrastructure and health services), communication media, and transportation, enabled the rapid, generalized, and massive therapeutic and prophylactic use of antimicrobials in hospitals and community settings. World Wars (WW) occurring between the “first *globalization*” (second half of the 19th century to 1910) and “the second globalization” (1945 to 1990) also triggered the development of logistics to control infectious diseases epidemics and illness in international campaigns, which included mass drug administration of antibiotics and antiseptics to troops [16]. The serendipitous discovery of the effects of antibiotic byproducts (vitamin B12 coproduced with streptomycin, aureomycin derived from tetracycline) on animal growth by the early 1940s drastically changed the farming sector and food industry and widened the big pharma business sector [17,18,24,25]. By the end of WWII, the global human population had directly and/or indirectly been exposed to diverse antimicrobials and antiseptics for decades [15,16]. Soon after, the global spread of multidrug-resistant (MDR) organisms was a reality [26,27].

The second (1945–1989) and the third globalization (1989–2008) waves reflect two major global socioeconomic timeframes that greatly impacted the healthcare system and AMR in different ways. The bimodal economy models established after WWII and represented by the free markets from Europe and America, and the centralized planning markets from the Soviet Union and China resulted in major differences between the health sectors (and other industrial sectors) of the countries aligned with these two models. The 1978 Alma-Ata declaration’s vision for societal health pinpointed the need to reorientate health systems toward primary care to address the social and environmental determinants of health and inequality [28]. The Alma-Ata declaration used the definition of *health* given in the 1948 United Nations declaration [29], which added a political dimension and the need to involve various sectors to counteract health inequalities (a change from the traditional vertical public health perspective towards horizontal “health promotion”). However, this interpretation of health has not been fully understood by major health stakeholders [28]. The end of the Cold War and the birth of the World Trade Organization in the 1990s (with the late inclusion of China in 2002) meant that the breakdown of national barriers to trade and migration, the generalization of international traveling, and the implementation of the internet provided unique opportunities for global expansion of hosts, food, goods, and microbes [14,30,31,32,33]. In 2005, the Organization for Economic Co-operation and Development (OECD) alerted the public to the involvement of economic growth sectors in the problem of AMR and the need for intersectoral cooperation at a global level to fight the AMR problem [34]. A few years later, the “O’Neill reports” highlighted how AMR could influence emergent economies, grouped under the terms BRICS (Brazil, Russia, India, China, and South Africa) and MINT (Mexico, Indonesia, Nigeria, and Turkey) [10,35].

The fourth wave of globalization (the Fourth Industrial Revolution, 4IR) brought novel challenges, such as the rapid emergence of ecological threats, the rising of the absolute number of the human population (with a global increase in the elderly), the increased *social atomization*, the political polarization, and social inequality [36]. In 2018, the Astana declaration (Alma-Ata 2.0) emphasized once more the need to reorient health systems toward primary care to provide universal health coverage instead of hospital-focused systems or low investment in health [28]. The novelty of the Astana declaration was to link health with the achievement of sustainable development goals (SDGs) [28]. AMR is also linked to SDGs by the quadripartite World Health Organization (WHO)–Food and Agriculture Organization of the United Nations (FAO)–World Organization for Animal Health (OIE)–United Nations Environmental Programme (UNEP) [37,38].

Other than inequalities, the current *social atomization* also impacts the increasing knowledge fragmentation, the loss of a comprehensive vision of the sociocultural frame, and the rapid obsolescence of social habits that greatly influence the vision and the way to approach health and scientific questions [36,39,40].

### 1.2. The Understanding of AMR in the Healthcare Network: From Disease Ecology to Evolutionary Biology and Social Sciences

AMR has been analyzed through the lens of different disciplines and conceptual frames. The belief that infectious diseases could be eradicated by novel “*miracle*” *drugs* was soon replaced by increasing evidence that unnecessary antibiotic use contributes to the evolution of infectious pathogens. This warning, originally raised in the mid-1940s by René Dubos [41,42], contributed to the ecological understanding of infectious diseases’ causation, now adding chemotherapeutic agents and technological advances of modern medicine to other environmental factors that have forced bacterial adaptation. Dubos was the first to conceive of microbial evolution within a large-scale and long-term scenario and greatly influenced clinical microbiologists and infectious disease physicians after the publication of his seminal book, *The Mirage of Health* [43]. They started to apply disease ecology principles to short-term and local events derived from AMR in hospitals and farms. Early data collection noted a change in the causal agents of hospital infections, now commensal opportunistic pathogens, such as staphylococci, Gram-negative microorganisms, and fungi [44,45], which often become resistant to multiple antibiotics. Early studies have also shown the vulnerability of hospitalized infected patients [45,46,47] and technology’s role in creating novel ecological niches that favor the unforeseen and abrupt emergence of infections [47,48]. Adopting the disease ecology perspective for infectious diseases led to hospitals being considered as environments and patients as structured populations. Standardization of the hospital infrastructure and medical practices during these early years in the US constituted the first step of current “evidence-based medicine”. However, these standards would not be employed by all medical practices or implemented at a global level until the 1980s and 1990s [2].

The swift escalation of antibiotic-resistant clinical isolates, especially after the global epidemics of penicillin-resistant *Staphylococcus aureus* in the 1950s, followed by a continuing endemic situation, motivated the arrival of public health-based and population-based principles to improve hospital hygiene performance in the United Kingdom and the United States by the mid-1960s [49]. Hospital hygiene became a discipline, and large hospital-acquired infection (HAI) control programs based on a risk-factor analysis and antibiotic stewardship policies (restriction of antibiotic use and treatment based on laboratory susceptibility information) were implemented [50,51,52]. More importantly, it meant that hospitals were defined as communities in which public health principles could be used to prevent and control HAIs [52].

By the late 1970s, the medical community had realized how rapid bacterial adaptation could occur through *plasmids* as vectors of antibiotic resistance genes (ARGs), enabling AMR to cross species and borders [18,26,27]. Although AMR started to be formally discussed at WHO during the mid 1970s [53,54], what placed AMR in the global scene were the initiatives led by Stuart Levy a few years later. At the closing session of the “Molecular Biology, Pathogenicity, and Ecology of Bacterial Plasmids” conference he organized in Santo Domingo, Dominican Republic, in 1981, over 200 clinicians and scientists from 27 countries signed a joint “Statement Regarding Worldwide Antibiotic Misuse” where AMR was firstly categorized as a “worldwide public health problem” and awareness of the consequences of antibiotic misuse at all levels of usage were firstly highlighted [55]. This seminal statement led to the foundation of the Alliance for the Prudent Use of Antibiotics (APUA), a global nonprofit organization with the common concern and vision of educating the medical profession and public alike about appropriate antibiotic use, promoting the global uniformity of regulation regarding antibiotic sales and labeling, and turning attention to the need for more standardized and coordinated surveillance of AMR [56]. Then, AMR began to be defined as a “global” (pandemic) and “ecological” problem [33,57,58]. Moreover, at a conference in Paris in 1988, Joshua Lederberg first included the evolution of AMR in a new category he named “emergent diseases” [59], in part to recognize and recall the landmark contribution of René Dubos’s ideas to the fields of AMR and disease ecology. Although this categorization primarily highlighted the epidemiological connotations of AMR, it also referred to the rapidly evolving processes followed by long-term changes in bacterial ecology.

Until the 1990s, the control of infectious diseases had been prioritized over analyzing the biological evolution of microorganisms. During this decade, we witnessed the arrival of molecular epidemiology and “-omics” developments and the adoption of a population genetics lens to recognize the impact of various units of selection (multilevel *populations*) in the evolution of AMR [40,60]. These advances brought an explosion of descriptive genomic studies that exhaustively addressed the ubiquity and diversity of species and ARGs and also revealed the AMR transmission pathways between and within the One Health sectors with impact on the way to face the control and prevention of AMR [3,4,61,62]. Afterward, the analysis of AMR in microbiomes and the birth of the *resistome* concept [63,64] changed the way of defining and approaching AMR (Appendix A). Variations in microbiome composition were recently associated with the host’s susceptibility to antibiotics and to pathogen translocation/invasion [65,66]. Moreover, the application of an ecological framework is promising for understanding host–microbiome interactions at different scales (living and abiotic environments) and, thus, selection and transmission events [67]. However, it implies the revision of some of the *Henle-Koch’s postulates*, such as the causality of infectious diseases by a single “etiological organism” [68,69]. In a formal epidemiological sense, alterations of the microbiome, but also hereditary traits and immunity, act as confounding variables on the expression of an infectious organism.

Most of the studies published during the 1990s and 2000s were disconnected from the control of the disease, which led to the categorization of this time as “the obscure era for infectious diseases” [2,70], despite the great advance in basic knowledge they provided on the ecology and evolutionary pathways of microbial entities. In the 2010s, many voices claimed the inclusion of evolutionary biology as a basic science for medicine [71]. Nonetheless, ecology remains to be essential to understand the fundamentals of disease in the era of the microbiome [67,72,73,74]. Multi and transdisciplinary approaches taking into account social sciences are increasingly demanded to understand and face Global Health challenges, including AMR [13,75,76,77].

### 1.3. AMR at the Healthcare Network in the 21st Century: Novel Challenges in a Global World

Socioeconomic changes in the 20th century have contributed to abruptly modifying the fluxes of humans, animals, goods, and microbes that affect the emergence, transmission, and burden of AMR at the local, national, regional, and global levels [14,30,31,32,78]. Human migrations and/or international travel play an important role in the global spread of MDR bacterial strains of certain species, such as *Mycobacterium tuberculosis* [79], *Salmonella typhi* [80], or Enterobacteriaceae resistant to third-generation cephalosporins, carbapenemases, colistin, or fluoroquinolones, which are easily recognized when introduced into low-incidence countries [81,82,83]. Changes in social, individual, and collective habits influence issues, such as antibiotic use [12] or the emergence and transmission of novel threats (e.g., Shigella in HIV-negative men who have sex with men) [84,85]. Novel food habits have also triggered the mass production of food animals and the use of antibiotics in veterinary and affect the rates of AMR at a local level [86,87]. Demographic changes, including the rise in elderly and life expectancy rates, are predominantly affected by infections caused by MDR strains of commensal opportunistic pathogens [88]. Moreover, *pandemics* of both non-communicable and communicable diseases [77], emerging infectious diseases, and other global health challenges are now part of a colossal global *syndemic* landscape [75,89,90,91,92] in which the altered conditions of a particular ecosystem are readily (and synergistically) transmitted to others. The results of such interactive globality modify antimicrobial usage, infection control programs, and transmission pathways that affect the emergence, transmission, and burden of AMR in healthcare centers and the community [93,94,95]. Today, AMR disproportionally impacts novel disparate vulnerable populations (e.g., children, the elderly, hematologic-oncologic, immunocompromised, or undernourished populations, migrants, and citizens with inadequate sanitation infrastructures, or those in war zones) who require medical care, drug therapy, and/or chemoprophylaxis to prevent or control different infectious diseases but with disparate access to antibiotics and health coverage [94,96,97]. Climate change is predicted to increase the rate of human infections, particularly those caused by MDR pathogens [98,99,100].

In summary, interactions between heterogeneous microbial communities vary in non-additive ways that may influence pathogen invasion and persistence at different spatial levels [74,78]. The estimation of the relative contributions of *heterogeneities* across scales (hosts, pathogens, environmental-healthcare patches, systemic and institutional drivers) determining pathogen transmission and persistence [101] is a pending issue of *disease ecology* and, thus, for the adoption of appropriate measures to combat/control infectious diseases and AMR.

## 2. Reframing AMR and Infectious Diseases in a Global *Syndemic* Scenario

AMR has been a public health priority for decades. In 2019, the WHO included AMR as one of the top ten threats to global health [1]. Today, all international organizations stated that AMR should be approached from the perspectives of One-Health and Global Health [35,38,61]. However, the implementation of efficient measures in different socioeconomic and geopolitical timeframes remains a major challenge.

### 2.1. Public Health to Control AMR in a Global Syndemic Scenario

Public health principles have been applied to prevent and control HAIs and AMR in the healthcare network [52]. Surveillance, antibiotic stewardship, and hygiene control programs have been the cornerstones to mitigate these challenges. Surveillance shows limitations related to the difficulties of the diagnostics to validate endpoints (e.g., definition and interpretation of HAIs cases, the effect of underlying conditions, etc.) (see Section 3 and Section 4.1). The COVID-19 pandemic has fueled the AMR global crisis due to “the overuse of antibiotics to treat COVID-19 patients, disruptions to infection prevention and control practices in overwhelmed health systems, and diversion of human and financial resources away from monitoring and responding to AMR threats” [102]. Concurrent infectious diseases or pandemics would have similar effects [43,54,55]. This “novel” reality represents a change from the traditional vertical public health perspective focused on reducing the spread of single pathogens towards horizontal “health promotion”, also accounting for socioeconomic and environmental factors [3]. More transversal programs (coined as “Lateral Public Health”) to enhance the alerts between healthcare centers and community hotspots by surveillance of vulnerable populations in community health centers (CHCs) would help to meet challenges in the 4IR [75,103]. The influence of the socioeconomic and cultural context in behaviors on antimicrobial stewardship and use of healthcare systems (e.g., healthcare-seeking behavior, financial reimbursement systems, institutional quality management, governmental incentive systems, regional hospital network structures, and hospital referral practices) impact the implementation of public health measures and, thus, the prevalence and diversity of HAIs and the transmission of AMR pathogens. A scientific approach to explore and understand the influences of these factors is starting to be adopted [13,104,105,106].

### 2.2. AMR and Health, One-Health, and Global Health

The term “One Health” focuses on the risk assessment of AMR’s emergence, transmission, and maintenance at the interface between humans, animals, and their related environments. The One Heath sectors comprise *heterogeneous* populations with varying interaction intensities and levels of *agency* [61,78,107]. Various studies in Western countries have demonstrated distinguishable transmission cycles in each One-Health segment [4,108] with much less cross-sector impact [3,4,109,110]. More specifically, AMR transmission pathways predominantly occur within “*hothouses*” (e.g., hospitals or farms) or associated networks (e.g., primary-secondary-tertiary healthcare levels), which accumulate some heterogeneous and synergistic risk factors, such as crowding, selective pressure, vulnerable populations, environmental reservoirs of MDR pathogens, or transmission vectors, such as chronic patients or medical workers with high (inter) healthcare exposure. Wastewater treatment plants are increasingly recognized as AMR hotspots [38,111] (Figure 1).

The social and economic inequalities triggered the adoption of universal plans and policies under the “Global Health” umbrella [28,37,62]. However, the 20th century’s general improvements in health, such as care systems, infrastructures, or governance resulted in byproducts of social prosperity and inclusion (e.g., increased life expectancy and changes in the health care system focused on the elderly or on pediatrics) has given way in the 21st century to narrow approaches based on targeted interventions and narrow international aid (e.g., “mass drug administration programs”, support to war conflicts, and humanitarian health assistance). Moreover, the current global economic system prioritizes “wealth” over “health”. In fact, the effectiveness of interventions and analysis of AMR outcomes are usually expressed in cost by different official bodies, such as the Centers for Disease Control and Prevention (CDC), European Centre for Disease Control and Prevention (ECDC), OECD, WHO, World Bank, and social independent corporations, such as the ReACT group RAND Corporation (Research ANd Development) [8,34,90,97,115,116,117]. Again, this is probably influenced by the misinterpretation of the *health* definition that tends to indistinctly consider “disease care” and “health care” [91].

### 2.3. Ecosystems and Their Feedback Loops: The Environmental Dimensions of AMR

AMR control is heterogeneously influenced by the One Health sectors, their associated economic growth groups, and their loops (Figure 1 and Appendix A). The time of variation within each compartment or ecosystem is linked to changes in the behavior and culture of these human communities [113,114]. According to “the biological ecological model of human development”, the temporal scale of changes decreases from *microsystems* (such as interaction within families or friends) to medium-sized ecosystems (*mesosystems*, such as hospitals or workplaces), and larger ecosystems (*macrosystems*, mostly involving activities, such as immigration, tourism, or changes in the food or pharma industry) [113,114]. The “sustainability and management ecosystems” are based on the control of others (*metasystems*, including social norms, such as family status, professional/educational status, culture, behavior, cultural norms, beliefs, and religious practices), policies and interventions, and economic structure [118]. These ecosystems are not only linked by a hierarchical structure, but they continuously change—each dependent on the others—linking in-loop dynamics of health, business, governance, and ecological activities. Indeed, the outcomes of such dynamics are directed toward improving human and environmental health and well-being, scientific development, economic prosperity, and social equity. In all these ecosystem dimensions, AMR plays a significant role, and, in turn, all of them influence AMR’s evolution.

Other than the need to include the socioeconomic and cultural context in planning public health interventions mentioned above, the time dimension, barely considered in previous operational conceptual frames, will be a key factor for planning and decisionmaking. Muti-level ecological systems considering social traits, including age and family structure, space and time location of individuals, and social dynamics, in general, have been included in the computational simulation of AMR evolution [119].

## 3. Challenges in Measuring AMR in the Heterogeneous Healthcare Network

AMR information is mostly provided by hospital records that also feed local and regional surveillance systems. Correlation of these data with demographic information from whom the pathogens are isolated (to date, mostly humans and mostly inpatients) offers insight into the underlying epidemiology and facilitates the formulation of rational interventions aimed at reducing the burden of AMR (diagnosis, treatment, prevention) and its monitorization [56]. The available information on AMR came from relatively few sources using similar indicators, methodologies, and models for disparate purposes (“a one-for-all surveillance system”). Some studies have categorized the various surveillance programs according to their *scale, scope, and structure* [120,121,122,123,124]. However, a reflection on the *validity* and *reliability* of the AMR information to be applied in various healthcare (primary, secondary, tertiary), demographic, and economic contexts (local, regional, global, and inter-sectorial levels) has been neglected to date.

“Validity” (or “construct validity”) defines how well a test, indicator, or experiment measures up to its claims. It refers to whether the operational definition of a variable actually reflects the true theoretical meaning of a concept. In other words, does the indicator or the instrument measure what it is supposed to measure? “Construct validity” comprises all other types of validity evidence, such as *content validity*, *instrument validity,* and *criterion validity*. “Reliability” (or reproducibility) in statistics refers to consistency in measurement: the capacity to produce the same result for two identical states; or, more operationally, the closeness of the initial estimated value(s) to the subsequent estimated value(s) (see Section 3.3).

This section defines the *unit of analysis* and *indicators* used in AMR and discusses the factors that affect their validity and reliability (Figure 2).

### 3.1. The Sample (The Unit of Analysis)

Microbial infectious diseases and AMR disproportionally impact populations and settings, and the “who” or “what” that might occur needs analysis. The *units of analysis* in AMR are human groups (such as single individuals, social, professional, or health-risk groups), healthcare settings (tertiary, secondary, primary, or single wards), and economic sectors (One Health, economic growth sectors) [125].

Age, sex, body health condition, genetics, and the *exposome* influence host susceptibility, host immunity, and social behavior [113]. The classical high-risk populations prone to acquire AMR are individuals with severe underlying diseases, immunocompromised conditions (children and elderly), living in poor socioeconomic structural environments, or significantly exposed to factors that increase the risk of selecting or spreading AMR (e.g., invasive procedures). In low- and middle-income countries (LMICs), risk groups for AMR also include those enrolled in mass drug administration programs to prevent infectious diseases, such as children younger than five years (to prevent trachoma or group B streptococci infections), HIV contacts, or those affected by respiratory and enteric infections [125,126,127]. Some individuals can be responsible for a large number of secondary contacts (“superspreaders”), although the factors beyond this phenomenon remain unclear [128,129].

The risk of acquiring and spreading AMR also varies at different healthcare levels. Traditionally, populations are analyzed at tertiary and secondary healthcare centers (hospitals and long-term care facilities), where the concentration of the high-risk groups of people and online diagnostic and life-sustaining equipment (such as ventilators), indwelling catheterization, and frequent use of drugs facilitate the emergence and local evolution of AMR. Within hospitals, some wards concentrate on groups and factors at higher risk than others [130,131]. Primary healthcare is greatly influenced by structural, governance, and management networks [105,106,132]. AMR in health system loops and One Health sectors has recently begun to be analyzed [133,134] (See Section 3.2 and Section 4).

### 3.2. The Indicators

In AMR surveillance, *indicators* (also called Operational Units of Surveillance, OUSs) [120] refer to the percentage of pathogens resistant to a particular antibiotic (pathogen–antibiotic pairs) at particular sites; the proportion of AMR-HAIs cases in population-based studies as markers for significant emergent threats; and the density of antimicrobial use (AMU) and/or consumption (AMC) as drivers of AMR. Hospital laboratories identify the microorganisms of any clinical sample they receive and have access to hospital AMC/AMU information. Surveillance programs at local, regional, global, or sectorial levels are fed with the OUS’ information (*sentinel species*) from non-categorized patients in large hospitals and with AMU/AMC from selected institutions/departments [121,122,135].

*Sentinel microbial species*. Only a few bacterial and fungal species acquire and successfully spread ARGs of medical relevance, following the Pareto principle (80% of consequences come from 20% of causes), as occurs for causal agents of many infectious diseases [125]. These species constituted the “*watch lists*” or “*priority lists*” *of antibiotic threats* of the WHO and the CDC, which only include a few traditional causes of major “classical” human infectious diseases (*Neisseria gonorrhoeae*, *M. tuberculosis*, *Streptococcus group A*, and *Streptococcus pneumoniae*), all affecting specific risk groups of community-based people and often associated with *syndemic* processes affected by social structures. The lists also included some commensal opportunistic pathogens, such as Enterobacterales, enterococci, and staphylococci, or environmental organisms (*Pseudomonas aeruginosa* and *Acinetobacter baumannii*). Recently, the Division of Healthcare Quality Promotion (DHQP) at the CDC reported that more than one fifth of HAIs are non-*Legionella* water-related opportunistic pathogens and warned of a possible major underestimation of these numbers [136]. The DHQP comprised multidrug-resistant Enterobacterales already included in the above-mentioned watch lists and other increasingly reported opportunistic pathogens (e.g., *Elizabethkingia*, *Achromobacter*, and *Burkholderia*) able to cause significant hospital outbreaks [137]. What all these prioritization guidelines reflect is a healthcare landscape comprising multiple *patches* of microbial species and host populations and the neglected analysis of the “hospital-built environmental patches” (sinks, wastewaters) in the emergence, transmission, and persistence of AMR [138].

None of the available “watch lists” consider the heterogeneity within a species. Again (Pareto’s principle), only a limited number of specialized lineages within each species significantly contribute to hosting and transmitting AMR. These have been identified as high-risk clonal complexes (HiRCCs) [139,140,141] or “pandemic clones” [142], which are operational designations helping target precision and preemptive interventions (e.g., control hospital outbreaks and prevent infections in the individual). However, the increased percentage of healthy human carriers of HiRCCs [143,144] has raised the possibility that these populations (even before acquiring AMR) are a natural part of the commensal flora, which can be spread to relatives when the population size increases, and the shedding is high [145]. The increase in number and exposure to various hosts and environments necessarily tends toward an increase in internal genetic diversity, which includes the emergence of hybrid lineages, such as ST258 *Klebsiella pneumoniae*, ST239 *S. aureus* [146], or EHEC *Escherichia coli* [147], or “jumps” of AMR bacterial lineages between humans and non-human hosts [109]. Both diversity within clonal complexes and clonal fluctuations with periodic emergences of new genotypes enable a permanent bacterial diversity that makes it difficult to rank and predict the risks of critical sub-populations. Genomic typing to track AMR and outbreak investigations is progressively implemented in national and international surveillance programs [148]. It demands a high level of agreement between genotypes and phenotypes, which seem to work for the most common pathogens and antibiotics included in the WHO’s and CDC’s “watch lists” [149].

*Genes, plasmids, and mobile elements*. ARGs can be rapidly spread across bacterial species, genera, or microbial kingdoms of various human, animal, and environmental backgrounds through transposable elements, often embedded in plasmid entities [78,150,151,152]. Once again, only a few members of plasmid, transposon, integron, or insertion sequence families are responsible for the spread of ARGs [151,153,154]. However, its diversification (e.g., F plasmids in Enterobacterales) [153,155,156,157] and promiscuity (e.g., carbapenemase encoding genes and the mobile genetic elements in which they are located) [158,159] often result in mosaic elements that difficult the choice of diagnostic tools, the identification of transmission pathways, the implementation and monitoring of interventions to target AMR plasmids. There are plenty of unresolved questions that require a deeper understanding of the dynamics of ARGs and plasmids in microbiomes beyond the analysis of every single entity [160].

*Microbiomes and resistomes*. Although not included in any surveillance program, metagenomics-enabled surveillance methods offer the opportunity to improve the detection of both known and yet-to-emerge pathogens [161]. The analysis of the intestinal and respiratory microbiomes and the resistomes may also allow clarifying fundamental questions, such as the host’s susceptibility to infection, therapeutic regiments, fecal transplantation, long-term AMR colonization, or health response to antimicrobial disturbances [65,162,163,164,165]. Other relevant and unexplored aspects are related to the microbiome of the hospital-built environment, an area of increasing interest in research agendas [166,167]. Unmet questions include the distribution of microorganisms between hospitalized patients and the “ward/room environment”, or the impact of the “hospital-built environmental microbiome” on patient health and indoor and outdoor environmental contamination [66,168,169]. Knowledge in this area is still scarce, and the tools and instruments are insufficient for acquiring valid and reliable information [170,171].

*Antimicrobial consumption (AMC) and antimicrobial use (AMU)* reflect how antimicrobials are dispensed and prescribed. Although they are often used interchangeably, they measure different things. AMC reflects the volumes of antimicrobials distributed (e.g., at a hospital or pharmacy), and AMU tells us how antimicrobials are used at the patient level (e.g., what diseases/pathogens are targeted, routes of administration). AMC/AMU data are primarily relevant in combination with local AMR trends and the preparation of antibiotic stewardship guidelines. Protocols and tools to properly collect AMC/AMU information are currently being debated [130,131,132]. Nonetheless, AMU data are still unavailable in most countries and hospitals, and antimicrobial exposure is generally reflected as AMC [94]. Surprisingly, only a few guidelines clearly state which antibiotics should be routinely monitored. A consensus list based on the antibiotics most used or included in the “Watch” and “Reserve” categories of the WHO’s Essential Medicines List AWARE index has been suggested to harmonize information [131,132]. 

AMU/AMC variations and shifts at local levels (i.e., outpatient stewardship) and regional levels reflect cultural and socioeconomic differences or secondary public health problems [13,94]. For example, cephalosporins and broad-spectrum agents, such as fluoroquinolones, macrolides, and second-line agents, such as oxazolidinones, have increased in LMICs and decreased in high income countries (HICs). The dramatic increase in the use of cephalosporins is due not only to economic growth, but also to changing prescribing practices for respiratory tract infections, skin and soft tissue infections, gonococcal infections, and enteric fever, in which cephalosporins have replaced penicillins and quinolones for infection management due to rising resistance [90]. Such differences in AMC affect the selection, emergence, and further transmission of AMR between distant geographical areas. Counterfeit and substandard antibiotics represent up to one third of the available pharmaceuticals in LMICs (42% of all reports received by the WHO Global Surveillance and Monitoring System on substandard and falsified medicines worldwide came from Africa, most corresponding to antibiotics and antimalarials) [32,172]. Of special concern is the scope of hospital and outpatient drug stewardship programs because many non-antibiotic drugs have an antibiotic effect and can contribute to the selection of MDR (*bystander selection*) [163,173,174] (Figure 2).

**Figure 2 microorganisms-11-01050-f002:**
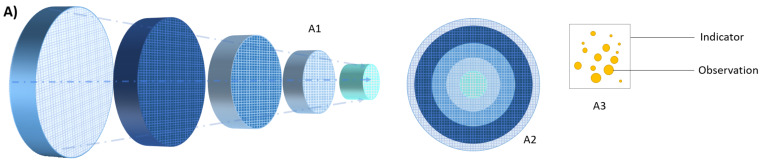
Challenges of AMR measurements. The validity, reliability, and comparability of the information [175,176,177,178,179,180,181,182].

### 3.3. Validity, Reliability, and Comparability of the Information

All AMR stakeholders use different indicators for their analysis. The pitfalls of multi-indicator analysis can be grouped into the categories of *scope, data quality, and comparability* (Figure 2). An evaluation of the suitability of multi-indicators in decision making has been neglected for the AMR field, although it is widely used in social sciences.

*Scope issues. The “validity” of the information.* The scope issues tell us the extent to which we are able to collect the appropriate indicators (*content validity*) to analyze various individuals, populations, or setting levels (*units of analysis*) and in a balanced and discriminated way (*instrument validity*). Although AMR surveillance employs different indicators/ OUSs to describe trends for a specific *unit of analysis* (individual patients, populations, healthcare institutions, and settings), traditional OUSs information is often too narrow to consider the full richness of such a context. AMR composite indicators are needed to analyze the population (e.g., resistome and microbiome indexes) [171,175], settings (e.g., “drug resistance index”) [176], or different sectors [133]. A clear prior conceptual understanding of the indicators is necessary to avoid redundancy (hierarchy), disbalance, or coverage bias. The ratio of sample size/number of indicators is a sensitive issue. The development of such indicators is still in its infancy.

*Data quality issues.* The *reliability* of the information relies on the quality of the underlying data. Variations in the unit of analysis, indicators, measurement methods, definitions (e.g., AMR, see Appendix A), guidelines (e.g., European Committee for Antimicrobial Susceptibility Testing vs. Clinical Laboratory Standards Institute antibiotic susceptibility breakpoints), and/or the application/use of an indicator greatly influence the quality of the information and preclude its comparability. For example, current estimations on AMR trends (in terms of the impact of AMC) or disease outcomes (HAI mortality predictions) came from mathematical models using epidemiological counts that do not consider disparities in the incidence or prevalence of infectious diseases or the antibiotics used [117]. Temporal series also influence data quality. Long-term series may be dotted with changes in definitions and guidelines that occur during the study (this is especially relevant when historical series should be analyzed or compared). This confusing trend might increase in the near future, given that new pharmacodynamic parameters will be considered to complement the minimal inhibitory concentration [183].

AMR information is frequently valid and reliable, but it is greatly biased by the structure of the dataset (*indicator coverage bias*). This bias often occurs for AMR in the health sector if the ecology or physiopathology of the sentinel organisms does not match that of the unit of analysis. Blood and cerebrospinal fluids are predominantly screened in AMR surveillance, not only because of their clinical relevance but because this choice prevents some of the inconsistencies that arise from differences in clinical cases, different sampling frames, or heterogeneous healthcare utilization. However, invasive isolates might not represent isolates of the same bacterial species producing other infections. In addition, variations in blood culture frequency (non-differential sampling vs. differential sampling, linked to diagnostic practices and the frequency with which blood cultures are taken) result in an increasing uncertainty when comparing resistance percentages between hospitals, CHCs, and other settings. On the other hand, severe infections are uncommonly reported in CHCs, such as LTCFs, where monitoring AMR usually uses urine or respiratory samples [184]. (See also sentinel organisms in Section 3.2).

Finally, the assignation of an observation to a particular class (*data attribution*) is one of the most important challenges in AMR and infectious diseases that has traditionally been approached by *microbial risk assessment* [178,180]. Such an association is not always obvious. Observations can be attributed to wrong classes because the most adequate class is still unknown or to multiple causes. Research on AMR makes inferences at the level of bigger aggregates, such as healthcare networks, countries, geographical areas, One Health sectors, or industry/economic sectors. Practical problems with MRA include identifying the number of observations needed (e.g., critical points in the transmission chain) and how to group them under these aggregates, which is problematic if they are cross-border [133,134].

*Comparability issues. The “structural” information.* The comparison of the *unit of analysis* should take into account (socioeconomic) *structural*, *governance*, and *specialization* differences, which are conceptually linked. Socioeconomic structural differences have been pointed out as affecting AMR [35,96,185,186]. The O’Neill report highlighted how AMR would affect LMICs, especially BRICS and MINT, a term grouping some countries according to their economies. However, they differ in their underlying socioeconomic structures that are not considered in any analysis [35,185].

Healthcare systems also have relevant semi-structural differences in terms of the structure and functioning of countries. Only the OECD statistics provide useful information for understanding the dynamics of the healthcare business and hospital systems in various countries; however, it is restricted to the last three decades [19]. The availability of beds per hospital normalized by 1000 persons, the most used indicator, revealed a broad variety of systems (e.g., a high density of small hospitals in Asian HICs vs. a small group of large hospitals in Western countries) with a major impact on the diffusion of medical technology and healthcare consumption. The impact of these structures has not been evaluated.

*Specialization/networking differences*. In addition to the structural and governance differences, the units of analyses can also differ due to e.g., patch dependency and feedback loops. Properties that define complex systems are *emergence*, *feedback*, and *adaptation.* Such properties have been analyzed in the context of “microbial complex systems” [7,151] but not in the context of “social or economic systems” that influence microbial fluxes. *Emergence* describes the properties of the system but cannot be inferred from the elements that comprise it (e.g., heterogeneity in AMR distribution can be conceptualized as an emergent property of the healthcare system, but also the pharma, food, transport, economic, and other systems). Feedback describes the effect of a change on further changes [134]. Finally, *adaptation* refers to the adjustments in behavior in response to intervention (e.g., restriction in antibiotic availability triggers the “black antibiotic market” and the trade of counterfeit, substandard, or old antibiotics in LMICs). Tools for approaching “complex public health systems” are still in their infancy [91,107,187] (Appendix A).

## 4. Heterogeneity of Stakeholders Involved in the Global Health Network: Activities, Objectives and Challenges

AMR research and policies have involved many stakeholders with disparate interests and objectives. Grundmann suggested a wise and interesting transversal frame to approach the problem and join the efforts of stakeholders with common objectives [120]. On this basis, he identified three groups of “targeted stakeholders”: (i) directly and immediately affected by adverse healthcare outcomes (patients and families, doctors, clinical microbiologists, drug prescribers, and drug dispensers); (ii) those affected by the social impact of healthcare outcomes (policymakers, politicians, public health workers, health insurance companies, and infection-control experts); and (iii) those with societal and corporate responsibility (scientists and researchers, pharma industry, funding agencies, global health, and research donors and sponsors). Table 1 shows the objectives, activities, and information sources of stakeholders at these levels. Gaps and challenges to achieve such objectives (in fact, to face the AMR problem) are highlighted using this framework. Some strategic aspects are discussed below.

### 4.1. Patient-Centered (Addressing Clinical Demands)

The hospital is the most important landscape to detect the emergence and facilitate the evolution of AMR. On average, HAIs occur in 7% and 10% of hospitalized patients in LMICs and HICs, respectively [188]. Different estimations consider the high rate of HAIs caused by AMR organisms [189] and predict rising mortality rates due to AMR [35]. Still, the risk of acquiring infections during hospitalization greatly varies between wards (5–10 times higher in ICUs than in general wards) [47], and between institutions [94,115,190]. The elucidation of the hospital indoor microbiome to the health and safety of the patients would help the detection of critical points and assessment of infection control and intervention strategies [191,192,193]. Quantification of individual risks according to the patient (e.g., microbiota composition and host factors) and the built environment (hospital abiotic reservoirs and ward metrics) are some of the current major challenges not fully addressed in surveillance programs [171,175,194].

One of the major challenges continues to be the definition and interpretation of HAIs cases because it undermines valid estimates of infection rates. Nosocomial acquisition is arbitrarily defined 48–72h after hospital admission, but it can also occur after the patient’s discharge, creating a flux of microbes between healthcare centers, and other community ecosystems [195,196]. The increase in health centers (including CHCs) hosting heterogeneous risk populations (e.g., elderly, migrants, the poor, and migrant agricultural workers) enlarged the health network and led to the development of the category “community-onset healthcare-associated infections” to accurately address the complexity of the acquisition/transmission pathways of AMR [196,197]. The increasing number of studies that demonstrate long-term colonization of discharged patients and households [165] confirms previous studies that suggested the transmission of AMR between healthcare and the community [195,198].

### 4.2. Population-Centered (Addressing Policy Demands)

Considerable investments in surveillance, design of guidelines, and policies to contain the AMR problem in the global arena have been conducted with limited success. In May 2015, the 68th World Health Assembly adopted the “global action plan on AMR” [62] and called on WHO member states to establish national action plans (NAPs) aimed at obtaining sustainable access to effective antibiotics. More than 85% of the world’s population lives in countries with developed or are in the process of developing NAPs-AMR although only very few NAPs in LMICs are based on a situational analysis. Recently, the environmental dimensions of AMR, neglected in the WHO-Global Action Plan, have been revised by the UNEP and supported by the WHO-FAO-OIE-UNEP quadripartite [38]. Other white papers by the World Bank [199], the OECD [97], and many other international organisms have identified gaps and challenges at various socioeconomic levels. A governance framework has been suggested, conceptualized as a cyclical process between the three governance areas: policy design, implementation tools, and monitoring and evaluation considering various domains and indicators to implement goals and challenges from various organizations [181].

One of the key objectives of the WHO global plan to fight AMR is to improve awareness and understanding of AMR through effective communication, education, and training [62]. Communication campaigns have played a critical role in the history of AMR, from enthusiastically encouraging the use of antibiotics among humans and animals in the 1930s, 1940s, and 1950s, to clearly recommending the prudent use of them in all NAPs nowadays. In early years, the press greatly contributed to the popularization of the “miracle” effect of antibiotics and stimulated its consumption and mass production. Two articles reflect such social effects. One, published in the New York Times in 1936, reported the successful recovery of President Roosevelt’s son after treatment for deadly sepsis with sulfa drugs [200] and stimulated the sales of the drug, which were sold at low cost and without prescription since the early 1930s [11]. Other lay press, published in The Time in 1941, echoed the “miracle” outcome of several infection cases treated by penicillin and the problem and cost of mass production of the drug. Such article triggered a response by the Royal Army Medical Corps and forced the British government to consider this issue a national emergency [201] Afterwards, the serendipitous discovery of the effects of antibiotic byproducts (vitamin B12 coproduced with streptomycin, aureomycin derived from tetracycline) on animal growth by the early 1940s drastically changed the farming sector and food industry and widened the big pharma business sector. In the 1950s, the Food and Drug Administration (FDA) greatly contributed to the encouragement of antibiotic use for growth promotion with massive communication and awareness campaigns. [17,18,24,25]. Such a position was defended by many scientists beyond the publication of the Swann report in 1969 [202]. Until the 1970s, no voices claimed the prudent use of antibiotics (see Section 1.2). Since 2015, the WHO has implemented an annual global awareness campaign called ‘Antibiotics: Handle with Care,’ which take place during the *World Antimicrobial Awareness Week (WAAW*). Results differ between countries, and sound recommendations for tailored campaigns adapted to various cultural models and sectors and involving experts in health communication and social marketing seems crucial [203].

### 4.3. Microbial-Centered (Addressing Novel and Old Infection-Control Needs and Community Demands)

Many fundamental scientific questions relevant to understanding the development, selection, transmission, and persistence of ARGs remain to be unraveled. The Joint Programming Initiative on Antimicrobial Resistance (JPIAMR), an international collaborative platform coordinating and funding research on AMR, provides a research and innovation framework for joint actions, outlining key areas that should be addressed, and providing guidance for countries to align their AMR research agendas at national and international levels [204]. The Scientific Research and Innovation Agenda (SRIA) of the JPIAMR adopted the One Health perspective and is included in the WHO Global Action Plan on AMR [62] as a recommendation for national research plans. More specifically, the SRIA defines priority topics through which coordinated research activities are translated into new and/or improved strategies to address AMR (development of new therapies, stewardship of new and existing antimicrobials, and strategies to monitor and prevent the selection and spread of AMR between humans, animals, and the environment). Gaps and challenges of specific AMR topics have been highlighted in various workshops supported by the JPIAMR in the last years [38,130,131,132,204,205], as well as in reports by other policymakers, such as UNEP [38,97,206].

Although a colossal advance in knowledge has been achieved in recent decades with the support of public and private funding, we face new challenges associated with the 4IR, such as extended life expectancy, economic growth of pharma and food industries (see Table 1) that led to *consumptogenic systems*, intensified livestock practices, exacerbation of diseases by climate change, and loss of biodiversity by environmental pollution. They will open novel research areas and will also require the commitment of economic and social sectors [38,206] (Table 1). Still, we have not established the fundamental knowledge to understand the selection, transmission and persistence of AMR pathogens, the dynamic of complex systems (e.g., microbial interkingdom interactions) [150] and the ecotoxicity of pharmaceuticals and drug mixtures on selection and biodiversity [38,207].

**Table 1 microorganisms-11-01050-t001:** Frame to approach the AMR problem according to stakeholders’ groups with common objectives. Gaps and challenges.

Level	Stakeholders	Objectives	Required Information	Scope	Information Source	Gaps and Challenges
Patient-centered (addressing clinical demands)	Patients (and their families), doctors, clinical microbiologists, prescribers, and drug dispensers (pharmacists), and infection control practitioners.	Improving patient treatment. Design and implement standard treatment guidelines and essential drug lists. Expected burden of disease (BoD) at any geo-administrative level (individual setting).	Fine-scale information of individual risks for infection, colonization, and/or expected treatment outcome. Fine-scale information at setting level (identification of risks areas).	OUSs ^1^ Composite indexes (DRI, microbiome indexes) AMU	Real-time collection of local and stratified patient and AMC/AMU data. Lateral Public Health [208]	Develop criteria to define HAIs cases, still based on 48–72 h admission time [195,196]. Microbiome precision medicine [161,194]. Develop composite indexes [171,175]. Combine AMR and AMC/AMU trends in categorized patient populations [122,130,131,132,209,210]. Understand microbial transmission (plasmid, clones, and microbial consortiums). Plasmid surveillance: criteria, tools, and databases [156,160,211,212]. Evaluate the impact of the indoor microbiome on the health and safety of patients [164]. Implement AMU interventions based on behavioral models [13,213].
Population-centered (addressing policy demands)	Health managers, infection control practitioners, public health experts, and health insurance companies. Politicians, policymakers, economists, and economic growth sectors.	To estimate the impact of AMR at national, regional, or international levels (trends and benchmarking). Identifying the leading causes for AMR emergence and spread. Quantifying AMC (pharmaco-epidemiology, sales).	Information aligned with validity, reliability, and comparability data from local, regional AMR surveillance networks.	Surveillance. OUS (list of priority pathogens). AMC (sales). Composite indexes.	Real-time collection of local and stratified patient data linked to local, regional, and/or national databases. Active population-based surveillance at local and regional levels	Identify population-level factors/groups linked to the emergence of AMR Granularity of the information to extrapolate estimates. Harmonize units of analysis and indicators (with appropriate corrector indexes. See Section 3 (data quality and comparability issues). Omic tools and infrastructures (capacity building, harmonized and standardized tools) [149,161]. Put AMR in the context of other Public Health threats (syndemics). Real-time surveillance only available in a few countries [121]. Community level surveillance adapted to healthcare loops (hospital wastewater, and insurance health networks,) [138,198,214].
		To design guidelines and policies. To monitor the effect of interventions.	Information aligned with validity, reliability and comparability data from regional AMR surveillance networks.	Aligned with National Antibiotic Plans (NAPs).	Active OneHealth sector-based surveillance aligned with NAPs	Harmonize NAPs according to local data and governance [181].
		Estimating the cost–benefit impact on public health, environment, and economy (BoD, biodiversity).	Return estimation ISOR. Social impact.	Public Health economics.	Public Health and Social data repositories.	Integrate the human, animal, and environmental policy outcomes with the economy markers.
		Efficient *awareness* tools and campaigns between stakeholders and lay public to inform alerts or interventions (control, drug policies,…).	Awareness through feedback influencing consumers, policies, and investment.	Awareness Tailored according to social and cultural norms.	Consumers-public polls and enquiries. Data from patient’s associations (ITUs).	Develop efficient communication tools and channels (intra and inter-sectorial) [203]. Assure updating of messages [203]. Revise prescription models.
		Revise *market and marketing* practices (pharma and food industry).	Evolution of sales and prescriptions. Control good practices.	Market and marketing.	National Health and Consumer services.	Strengthening governance, management, innovation, and financing [181].
Pathogen-centered (addressing infection-control demands) ^2^	Scientists and researchers, pharmaceutical industry, funding bodies, and global health donors.	Identification of transmission routes and microorganisms. Identify diagnostic biomarkers. Identification of reservoirs of MDR bacteria, plasmids, and hotspots for horizontal gene tranfer Identify PD parameters as minimal selective concentrations. Identify PD parameters as ecotoxicological concentrations.	Scientific publications, grey literature, workshop reports, international dossiers, white papers, and society reports.		Strains of microbial species (bacteria, fungi) of biomedical relevance from catalogued repositories and type strain collections.	Determine ethe effects of AMU (antibiotics and non-antibiotic antimicrobials) on dysbiosis patterns [164,215]. Understand microbial transmission (plasmid, clones, and microbial consortiums) Understand transmission pathways. (e.g., effects of particulate matters in the bacterial and fungal transmission, particularly in water–soil edges [216]. Plasmid surveillance: criteria, tools and databases [156,160,211,212]. Involve engineers in Public Health policies and guidelines. Environmental impact of sanitation measures [38]. Effects of interkingdom microbial interactions in the dynamics of MDR bacteria and ARGs [150]. Actions targeting the reduction of pharmaceuticals in the environment including the design, synthesis, and production of pharmaceuticals [38]. Determine the concentration of selective antibiotics for AMR in local environments [205,217,218]. Determine the impact of drugs in the environment to regulate microbial residue limits (MRLs) [218] ^3^.

^1^ The number of Operational Units of Surveillance (OUSs) tested greatly varies for different *units of analysis*. Although *routine clinical laboratories* measure all pathogens against a battery of antibiotics in order to guide therapy of individual patients, most surveillance studies are based on the collection of these data. ^2^ The increasing annual global growth of the pharma sector, prescriptions, and sales of pharmaceuticals at a rate greater than the increase of the population and with up to more than 50% of people in Western countries taking pharmaceuticals prescribed from more than one specialist, will force us to revise pharma workflow [38] as the effects of novel AMU/AMC patterns. ^3^ For most licensed medicines (+88%), data on environmental toxicity is not available [38].

## 5. Conclusions

AMR is one of the Global Health challenges of the 21st century that also threatens the achievement of several SDGs. The “*Great acceleration*” period has entirely transformed the health sector and health problems. While it led to the advance of human health, it also made humans increasingly vulnerable to contemporary challenges (e.g., AMR, emerging and re-emerging infectious diseases and non-communicable diseases), which are further impacted by climate change, poverty, conflict, migration, the increased *atomization* of the society, and knowledge fragmentation. Such a syndemic scenario showcases microbial communities in convergent vulnerable populations and disparate settings interacting with varying intensities that increase the risks to the health and safety of humans. The increasing lack of therapeutic options to treat MDR pathogens, the changes in the global drug market, the consequent decreasing interest of pharmaceutical companies in the development of new antimicrobials, and the spread of AMR across humans and animals worsen this scenario [18,48,219]. To date, the recommendations and guidelines related to the control of AMR in the healthcare sector rely on information provided by clinical laboratories and surveillance systems using similar indicators but poorly harmonized tools. The limitations about the validity, reliability and comparability of the information have been largely neglected and need to be revised in light of new challenges and taking into account methodological problems common to statistics in science under a multidisciplinary scientific framework. Public Health and Global Health policies and guidelines should be reframed to consider the novel problems in the 4IR and involve the global health network, including major economic sectors.

This review has only tried to bring together many aspects that are often considered separately. AMR is a public health problem resulting from social construction in the last centuries. Thus, the understanding of the history, the conceptual scientific frames, and the behavior of humans to face health problems are necessary to identify the current gaps and challenges and to resolve many already unresolved problems (the “known unknowns”).

## Figures and Tables

**Figure 1 microorganisms-11-01050-f001:**
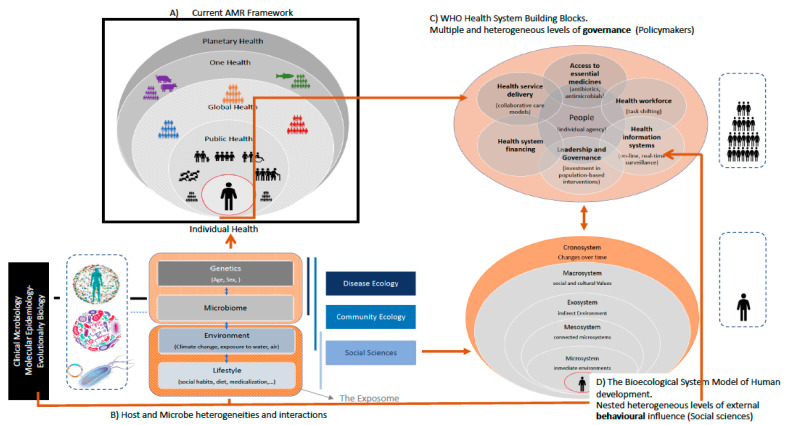
A framework for antimicrobial resistance in the healthcare network of the 21st century. A multidisciplinary approach for the analysis of AMR in a host metasystem landscape. (**A**) The current framework to approach health and global health challenges, which includes antimicrobial resistance (AMR) and pandemics [61]. (**B**) Multidisciplinary approaches to analyzing host and microbe heterogeneity and their interactions in the context of individual human health, which is influenced by intrinsic individual traits (e.g., genetics and physiology) and the exposome (exposure to environment/s, social habits, and contact with abiotic and non-abiotic entities). Individual microbial heterogeneity at sub-specific (gene, plasmid, clone) and supra-specific (microbiome) hierarchical levels is the focus of clinical microbiology and molecular epidemiology; host-microbe interactions and dynamics are analyzed by disease ecology and community ecology. (**C**) The WHO Health System Building Blocks framework, which was developed to promote a common understanding of the *health system* [112]. This is relevant for public health investments and results to feed global decision-making. (**D**) The Bioecological System Model of Human development. It establishes different levels (systems) of exposure to social groups [113,114]. These levels overlap those of microbial exposure. Orange-colored areas represent the influence of time in all systems (human, microbial, individual species, and institutions). Brown arrows represent connections between the various levels. Dotted boxes reflect the central targeted unit, namely humans, in (**A**,**D**); human groups in (**C**); and microbes and hosts in (**B**).

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
