# Peer review of "Antimicrobial Resistance in the Global Health Network: Known Unknowns and Challenges for Efficient Responses in the 21st Century"

_microorganisms, 2023, doi:10.3390/microorganisms11041050_

Round 1
Reviewer 1 Report
This is an interesting and thought-provoking review taking a “big picture” perspective on the issue of AMR and the ecosystem of emerging AMR threats.
The historical context provides an interesting background on the current situation regarding AMR threats and the hospital environment and would be of interest to those who don’t understand this. It also places the current situation in this historical context.
Overall, the text is quite long and I feel many of the concepts/sections could be synthesized to be a little less expansive.
While there is a section on AMC/AMU, and I understand the rationale for including this given the clear intersects between antibiotic use and resistance I wonder if this would be better if it was removed and the article focused on pathogen/AMR surveillance issues.
While the article focuses on health care settings, it is also very much talking about bigger picture “Public Health” There could be a better articulation of the difficulties of work in a “Public Health” context when working at the healthcare facility level.
In the end, I think there could be some better clarity on the “way forward” in the article – many of the complexities of the topic are presenting, but it would be good to highlight more clearly, if possible, the pathway for improving the situation in future.
Some specific comments that should be considered:
1. The abstract is broad in scope, and outlines many of the concepts in the paper. I think it would be worth trying to make this a bit more succinct
2. Throughout the paper, some of the phrasing / terminology / spelling could be improved - there are no line numbers to point to, but encourage a careful review of the whole text
3. There needs to be a figure (or possibly a table) that clearly defines what the key “unknowns” that need to move to the ‘known” category.
Author Response
Reviewer 1.
This is an interesting and thought-provoking review taking a “big picture” perspective on the issue of AMR and the ecosystem of emerging AMR threats.
The historical context provides interesting background on the current situation regarding AMR threats and the hospital environment and would be of interest to those who don’t understand this. It also places the current situation in this historical context. Overall, the text is quite long and I feel many of the concepts/sections could be synthesized to be a little less expansive.
Author´s response. The presentation of the manuscript has been deeply modified. Redundancies were eliminated and novel sections have been added, including a table and a figure. The abstract has been rewritten to reflect the current version of the work. We hope this revised manuscript addresses the demands of all reviewers.
While there is a section on AMC/AMU, and I understand the rationale for including this given the clear intersects between antibiotic use and resistance I wonder if this would be better if it was removed and the article focused on pathogen/AMR surveillance issues.
Author´s response. Surveillance of AMR threats and antibiotic stewardship are the cornerstones of AMR control. AMU/AMC are one of the main indicators or “operational units of surveillance” and are mentioned in different parts of the text. Thus, we consider that this part is essential to understand the problem in the context of this paper. This AMU/AMC description has been included in a novel section about the “AMR information” (section 3). We hope this will make sense to the reviewer.
While the article focuses on health care settings, it is also very much talking about the bigger picture “Public Health” There could be a better articulation of the difficulties of working in a “Public Health” context when working at the healthcare facility level.
Author´s response. We acknowledge this reviewer´s comment which is also aligned with one of the suggestions by Reviewer #3. The manuscript has been reoriented to make it useful to all policymakers with an emphasis on those working on Public Health.
In the end, I think there could be some better clarity on the “way forward” in the article – many of the complexities of the topic are presenting, but it would be good to highlight more clearly, if possible, the pathway for improving the situation in the future.
Author´s response. A table with gaps and challenges that need to be addressed in the future has been included (current Table 1). Also, Figure 2 offers some pitfalls of the information collected by surveillance systems.
Some specific comments that should be considered:
- The abstract is broad in scope, and outlines many of the concepts in the paper. I think it would be worth trying to make this a bit more succinct
Author´s response. The abstract has been deeply revised to reflect current contents and follow the reviewer´s suggestions.
- Throughout the paper, some of the phrasing/terminology / spelling could be improved - there are no line numbers to point to but encourage a careful review of the whole text
Author´s response. The manuscript has been deeply revised by professional English proofreaders.
There needs to be a figure (or possibly a table) that clearly defines the key “unknowns” that need to move to the ‘known” category.
Author´s response. A table of gaps and challenges has been included (current Table 1).
Reviewer 2 Report
This is an important review of the ecology of AMR in the healthcare context. The author provide a holistic view on the topic and bringing together conceptualizations from microbiology epidemiology and ecology.
As such the paper is timely and important
Several areas which could be further improved:
1. The business of MGEs is mentioned but the paper does not explore mobile elements as a "unit of outbreak" but rather focus more on clones. This could be further elaborated
2. The importance of the microbiome is increasingly being noted in the context of AMR but this is only very briefly mentioned and should be further enhanced. Areas of focus can include patient microbiota and ARB acquisition, ARGs harboured in the microbiome and the interplay with the environmental microbiota
3. The use of metagenomics and how it could better our understanding of the ecology of AMR
Author Response
Reviewer 2. This is an important review of the ecology of AMR in the healthcare context. The author provides a holistic view of the topic and brings together conceptualizations from microbiology epidemiology and ecology. As such the paper is timely and important
Several areas could be further improved:
- The business of MGEs is mentioned but the paper does not explore mobile elements as a "unit of outbreak" but rather focus more on clones. This could be further elaborated
Author´s response. Genes, MGEs, resistomes are discussed in the novel section 3. This aspect is not comprehensively discussed because the target audience of this paper is broad. Nonetheless, we address the gaps and challenges to consider them as an “indicator”. We hope to have addressed the reviewer´s demand.
- The importance of the microbiome is increasingly being noted in the context of AMR but this is only very briefly mentioned and should be further enhanced. Areas of focus can include patient microbiota and ARB acquisition, ARGs harboured in the microbiome and the interplay with the environmental microbiota.
Author´s response. The microbiome is now mentioned in different parts of the text. This issue is also included as hot topic in gaps and challenges (Table 1 and Figure 2).
- The use of metagenomics and how it could better our understanding of the ecology of AMR.
Author´s response. The use of metagenomics to better understand the ecology of AMR is mentioned in different parts of the text. Gaps and challenges of metagenomic approaches are mentioned. As mentioned before, the targeted audience of the paper is broad so details of interest for clinical microbiologists or molecular epidemiologists are not given.

Reviewer 3 Report
pt ID: microorganisms-1941860
Type of manuscript: Review
Title: Antibiotic Resistance in Healthcare institutions: Known Unknowns and
Implementation of Efficient Public Health Responses in the 21st Century
Authors: Teresa M. Coque *, Rafael Cantón, Ana Elena Pérez-Cobas, Miguel D.
Fernández-de-Bobadilla, Fernando Baquero
The MS provides an account of the multiple facets inherent to the emergence of antibiotic/antimicrobial resistance (AMR) in bacterial pathogens causing disease in humans. It addresses the gaps in the current scientific understanding of emergence and explores the consequential void in the available intervention repertoire. Thereby, a grandiose panorama of facts offer insights into the heterogeneities that drive AMR and our knowledge thereof, at all levels, historical, geographical, ecological, socio-economical, and evolutionary. This tour-de-force ends with a conceptional framework illustrated by a figure and a glossary meant to clarify the terminology introduced by the text.
General comments
The sheer scale of the task is Trojan and the result is a very demanding and ambitious MS. The title suggests that readers will gain an insight into the current knowledge gaps and strategies that should be implemented as efficient public health interventions. I wonder if the authors have been able to live up to the title’s claim. Rather than carving out and highlighting the unknowns (and portraying efficient public health interventions), the authors elaborate about what is known and how the extant multifaceted knowledge transcends different scientific domains. This often leads to lengthy accounts about interrelatedness of multiple phenomena at evolutionary, ecological and indeed socio-economical interfaces at every level of microbial habitats, from host to hospital wards to communities – and all at different geo-temporal dimensions. This results in long entangled sentences, with avoidable redundancies, limited scope and bewildering syntax. Thus, the text conceals rather than pronounces its main findings. The text would clearly benefit from a more concise elaboration of the main messages with some consideration for the reader by improving readability and lucidity. Regarding the reader, I was wondering: which audience is the text intending to address? Microbiologists, public health experts, politicians or scientific-minded lay-audiences? Clearly, when discussing the metrics of AMR quantification at the latest, all non-experts will have closed the screen. Would it not be important to address the non-experts? Otherwise, this effort follows the conventional pattern of medical microbiologists talking to medical microbiologists – and this communication bubble we keep cherishing for over 50 years.
Specific comments and remarks
Part 1. and 1.1.
This provides a historic account of the emergence and perception of AMR as an epidemic public health problem. The chapter ends with the notion that, thanks to advances in molecular epidemiology, the scope of the problem has been widened to include multiple bacterial species but also subgenomic genetic information in the form of mobile genetic elements (MGEs), without drawing conclusions on novel evidence-based interventions.
Page 2, line 1, title: consider changing “beyond” for “after”
“The Great Acceleration Period” One might guess that this encompasses the period of improving institutional health care from the likes of Nightingale/ Lister /Ogston until the great postwar bonanza of pills and profits when Pharmaceutical Industry-driven research started to market the majority of active compounds (which are still in use today) in the late 1950’s. Maybe this should be explained in the Glossary.
It would be useful for the reader to widen the Glossary at the end of the text and to include key words and concepts such as “death sinks”, “hothouses”, “consumptogenic systems”, “patches”, “hybrid lineages”, “orthogonality”, “abiotic reservoirs”, “market failure”, “evolution on a leash theory”. In addition, it would be helpful to add a parenthesis (see glossary) after the terms that have been included for explanatory purposes.
page 2, line 5: “healthy environments” considering the large scale outbreaks of infective jaundice (serum hepatitis) in the 1950’s in the US this may be a bit euphemistic.
page 2, line 20: replace “creating” by “helping”
Part 1.2.widens the view beyond health care institutions to include living conditions outside hospitals in an increasingly interconnected world.
What I am missing here are a couple of examples that would enlighten the reader towards a better understanding of the arguments put forward. I also miss examples that show the “… abruptly and synergistically changed AMR bacterial species … “ (lines 24-25). The text still focuses (lines 30-32) on AMR in health care centers, although there are ample examples of community pathogens with increasing and relevant AMR phenotypes such as M. tuberculosis, Shigella spp. and S. typhi.
The chapter also gives little credit to the extent to which health care seeking behavior and individual or collective agency are driving forces of current trends. It demands the need for knowledge about disease ecology which should inform “… appropriate measures to combat / control infectious diseases and AMR…” but there are no real world examples given.
Page 3, line 49: Consider revising “This Review discuss how AMR heterogeneity is measure…”
Part 2. and 2.1. describes efforts of Public Health Agencies to focus attention and control efforts to predominant target organisms using so-called “watch lists”. It introduces the important observation that the burden caused by AMR in hospitals is “Pareto distributed” or when using mathematical terminology obeys power law dynamics. This pervasive symmetry cannot be emphasized enough, as it applies at every levels of genetic abundance i.e. Species, Clones and MGEs and is the crucial ingredient of all hierarchically distributed self-organizing networks. It argues (lines 44-45) “The pandemic concept implies the widespread of the clones in the community” Apart from questionable syntax; I believe that this assertion is frankly wrong. Large-scale sampling and sequencing efforts across hospital and community boundaries have repeatedly demonstrated oligoclonality among the AMR isolates and high diversity among non-AMR carriage isolates.
This entire chapter would clearly benefit from a through revision especially linguistically. Some sentences are far too long (lines 4-13) and the semantics are easily lost to the reader. Again the authors make the point that targeted interventions, given the complexity of AMR behavior are difficult to conceive but do not suggest implementations of any kind as promised in the title.
Page 4, line 47: “hybrid lineages” are the result of large chromosomal replacements – maybe this should be mentioned?
Page 4, lines 49 -51: Sentence needs to be explained.
Part 2.2. purports to describe the heterogeneities among host populations who harbor AMR-pathogens or are more prone to infections and repeats to a certain extent what has been already mentioned under 1.2. Interestingly, it is argued, that the ecological disease understanding would invalidate Koch’s postulates in the light of the role of microbiota/microbiomes that may be conducive to disease expression. However, the same would hold for individual genetic host determinants as well as to immunity, both not mentioned in this paragraph. In a formal epidemiological sense, one would argue that alterations of the microbiome, hereditary traits as well as immunity might act as confounding variables on the expression of an infectious disease causally associated with specific pathogens.
Page 5, line 28: Explain why “convenience” – would social network analysis be the key word here?
Page 5, lines 31 – 34: Sentence should be shortened. Why “AMR in Western health care centers” would that not apply to low resource settings as well?
Page 5, line 42: Superspreading is by itself equally “Pareto distributed”, so you may want to leave out “while other not”
Page 5, lines 45 – 52: The sentences pick up concepts of potential interventions but not in a constructive manner (how could an efficient public health response look like?). The second sentence is rather deconstructive, portraying the dangers of broad-brush interventions in LMICs. Explain: GBS = Group B streptococci?
Page 6, lines 1 – 4: revise end of the sentence
Page 6, lines 5 – 8: This is rather a truism. Again no mention of how this knowledge could lead to effective public health interventions.
Part 2.3. elucidates the spatial heterogeneities and describes the process of dispersal from sources inside health care networks. This largely repeats notions already expressed in chapter 2.2.
Page 6, lines 13 – 14: “… following not only stochastic, but also deterministic processes… “ I don’t understand this. I thought that a deterministic process is the idealization (i.e. median behavior) of naturally occurring stochasticity?
Page 6, lines 14 – 20: I wonder, if this could be described in a bit more lucid manner?
The text also introduces the limitations of extant surveillance systems that focus on certain types of hospitals and indicator pathogens and describes the difficulties to extrapolate these findings for community action.
Page 6, lines 25 – 31: I find it difficult to believe that surveillance efforts that aim to quantify health determinants such as AMR, are designed to “tackle” disease. Foremost, they represent an early step towards measuring the size of a public health problem.
Page 6, lines 34 – 36: There are of course structural barriers to measure reality. For example, how would one go about measuring AMR in a Mumbai slum or in rural Tanzania?
I believe the biggest bias arises from the selection of patients for diagnosis. Diagnostic habits for many reasons are difficult to factor-in into the estimates. But this is more a technical issue and not caused by governmental interference: Governments before Covid found it very difficult to grasp the concept of measuring disease, at all. For vertical vs horizontal health interventions – see also the WHO Alma Ata Declaration of 1978.
Part 3. reiterates the difficulties when quantifying AMR as a public health problem.
Page 6, line 51 – page 7, line 2: I would like to see the assertion that “… surveillance has played a pivotal role in early alerting of microbial threats, identifying transmission within and between health care centers, and monitoring and guiding the impact of different interventions… “ referenced. My experience is that surveillance systems are notoriously too sluggish and cannot be utilized for early warning and response, neither can the remaining claims be supported by experience and published sources. Microbial threats (such as novel ARGs) have usually been identified in clinical settings, when treatment failed and good diagnostic laboratories where at hand. Transmissions are usually ascertained during outbreak investigations or purpose-designed structured surveys, and success of interventions can hardly be discerned at the high aggregations level of the data that are used for surveillance purposes. There may be exceptions such as interrupted time series analyses, but usually these are deployed during intervention (quasi-experimental) trials and are not part of larger surveillance activities.
Page 7, line 3 – 7: “Gaps and challenges….” This sentence contradicts the first sentence “… requires a cost-effective and targeted sampling strategy.”
Part 3.1. The term Operational Unit of Surveillance (OUS) also within the three mentioned contexts (AMR, AMU and HAI) was first introduced by [PMID: 24694024; Ups J Med Sci. 2014 May;119(2):87-95. doi: 10.3109/03009734.2014.904458. Epub 2014 Apr 3.] and this should be referenced.
Part 3.1.1. describes the tenet of international surveillance initiatives such as ECDC’s EARS-Net, ReLAVRA from PAHO/WHO and the WHO GLASS initiative that determines AMR prevalence on the basis of indicator drug-bug combinations in isolates from bloodstream infections and discusses the threats to validity of data generated in this fashion.
Page 7, lines 19 – 21: Sentence required clarification.
This chapter raises the issue of valid metrics (clinical breakpoints vs ECOFFs) for determining the prevalence of AMR as well the issue for the “correct?” endpoints. Although the text mentions proportion resistance among indicator organisms and clinical endpoints such as cause-specific morbidity and mortality as potential metrics, it however shies away from including a satisfactory discussion about the merits and complexities when choosing different endpoints for the sake of valid and comparable data generation by surveillance systems.
Page 7, line 30: Heavy metals are at present not used, neither for therapeutics (except for Leishmaniasis, and Trypanosomiasis) nor in infection control interventions.
The remaining text about AMR surveillance discusses the merits of including invasive isolates vs isolates from other sources. Rightly, the text argues that sampling (diagnostic habits) largely bias results and threatens the comparability across institutions as well as countries. Here the textbook example for the “death sin” in epidemiology is differential misclassification, when blood culture diagnostics is carried out as a last resort (when all treatment options have failed) and predictably, the surviving bacteria are likely to be of a MDR phenotype. Other sources of isolates such as screening isolates, environmental isolates as well as respiratory, intestinal and entire microbiomes are, to my knowledge, not included in surveillance systems. There are good reasons for the exclusion, but the text leaves a thoughtful discussion about the feasibility of these approaches to be desired for.
The next section mentions the difficulties when measuring AMC/AMU with respect to stewardship interventions, especially in LMICs where AMU is confounded by the marketing of counterfeit and substandard compounds. This is altogether an important issue but requires a more detailed consideration which is beyond the remit of the current text. This should probably be mentioned to the reader.
The same applies to the non-exhaustive nature of the discussion about surveillance of hospital-/health care-acquired infections (HAI). Here the text does not give sufficient room to the complexities of HAI surveillance. This issue merits more detail and a more thoughtful discussion about the limitations, validity of endpoints as well confounding (such as underlying disease of patients treated in hospitals, which is not mentioned at all). Mentioning of local ecological and socioeconomic factors is simply not sufficient to describe differential health care seeking behavior, financial imbursement systems, institutional quality management and governmental incentive systems, regional hospital network structures and hospital referral practices - all of which are impacting on the prevalence, diversity and composition of AMR pathogens in hospitals. Moreover, the biggest problem of HAI surveillance is how to define a nosocomial infection (including the arbitrary 48 hrs. rule, line 31), and the way in which case definitions and their interpretations undermine valid estimates of infection rates.
Part 3.2. The argument that surveillance of environmental hotspots is not harmonized misses the point. There are innumerable technical and operational issues that need to be overcome and, to my knowledge, such surveillance systems do not exist. Finally, the most important question would be what would be the benefit of a surveillance of environmental hotspots for those that need to know? Are other risks in the environment, chemical, radiation, etc. not more pertinent issues to map? More important are initiatives that aim at community pathogens especially in LMICs and the text is right to mention this dearth but should include Shigella spp. and S. typhi.
Part 3.3. Design of surveillance systems (including time-sensitivity of data) depends on stakeholder demands. What is needed by those who need to know? And what interventions could be informed by what kind of data? Are these time sensitive? A one-for-all surveillance system should be discouraged. I am missing a thorough discussion about the aims and objectives of AMR surveillance systems [see also PMID: 24694024; Ups J Med Sci. 2014 May;119(2):87-95. doi: 10.3109/03009734.2014.904458. Epub 2014 Apr 3.].
Part 3.4. This is more of the same and in the context of what has been discussed before a rather redundant enumeration of items. What I am missing are constructive suggestions for the utilization of surveillance data towards informed interventions.
Page 10, line 32, Title: Improve language. How does availability relate to AMR heterogeneities?
Page 10, line 43 – 44: What does triangulation mean in this context?
Page 10, line 49: Consider replaceing “scope” with rate.
Page 11, line 1 – 3: Sentences need to be revised.
Part 4. This chapter remains rather vague about the actual forces that shape the interactions that are describes in a very verbious manner.
Page 11, line 7 – 8. Consider rephrasing the first sentence.
Page 11, lines 17 – 19: Please explain what you mean.
Page 11, line 25: What precisely are ecosystem dimensions?
Part 5. and 5.1. Again, the text remains rather circumstantial without providing real world examples. It is stated, that interfaces are difficult to establish between human, animal and related environments, but the authors miss to mention studies that have tried to exactly do that – establishing the link between these OneHealth segments (using whole genome sequencing) and have quantified the net flux of AMR between these compartments. The authors also fail to mention that these studies (more often than not) were unsuccessful to establish genomic exchange across these interfaces. And therefore the important scepsis about OnHealth syndemic senarios is missing, rather the text recommends that “A multidisciplinary approach for the analysis of microbial metacommunities in a host metasystem landscape taking into account disparities across countries” needs to be established. There is no vision or idea mentioned on how this should or could be put into practice.
Page 13, lines 1 – 2: Revise sentence.
Page 13, lines 11 – 14: Revise sentence. Explain how “maket failure” adds to the problem.
Part 5.2. Much of what is written, becomes already clear from the previous text. The dichotomy of vertical public health approaches vs. horizontal “health promotion” was subject of the Alma Ata Declaration of WHO in 1978 and probably should be mentioned in this context. Some sentences lack logic.
Page 13, lines 25 – 27: Clarify the semantics of this sentence, please.
Page 13, 38 – 42: Improve the definition of the property of emergence “… as a phenomenon brought about by properties of a system, but cannot per se be predicted from the elements the system is made of…”
Part 6. Conclusions
Revise ordinal number in title.
Page 14, line7: Change “on” for over.

Author Response
Reviewer 3.
The MS provides an account of the multiple facets inherent to the emergence of antibiotic/antimicrobial resistance (AMR) in bacterial pathogens causing disease in humans. It addresses the gaps in the current scientific understanding of the emergence and explores the consequential void in the available intervention repertoire. Thereby, a grandiose panorama of facts offer insights into the heterogeneities that drive AMR and our knowledge thereof, at all levels, historical, geographical, ecological, socio-economical, and evolutionary. This tour-de-force ends with a conceptional framework illustrated by a figure and a glossary meant to clarify the terminology introduced by the text.
General comments
The sheer scale of the task is Trojan and the result is a very demanding and ambitious MS. The title suggests that readers will gain an insight into the current knowledge gaps and strategies that should be implemented as efficient public health interventions. I wonder if the authors have been able to live up to the title’s claim. Rather than carving out and highlighting the unknowns (and portraying efficient public health interventions), the authors elaborate about what is known and how the extant multifaceted knowledge transcends different scientific domains. This often leads to lengthy accounts about interrelatedness of multiple phenomena at evolutionary, ecological and indeed socio-economical interfaces at every level of microbial habitats, from host to hospital wards to communities – and all at different geo-temporal dimensions. This results in long entangled sentences, with avoidable redundancies, limited scope and bewildering syntax. Thus, the text conceals rather than pronounces its main findings. The text would clearly benefit from a more concise elaboration of the main messages with some consideration for the reader by improving readability and lucidity.
Regarding the reader, I was wondering: which audience is the text intending to address? Microbiologists, public health experts, politicians or scientific-minded lay-audiences? Clearly, when discussing the metrics of AMR quantification at the latest, all non-experts will have closed the screen. Would it not be important to address the non-experts? Otherwise, this effort follows the conventional pattern of medical microbiologists talking to medical microbiologists – and this communication bubble we keep cherishing for over 50 years.
Author´s response. We greatly thank the reviewer for his/her exhaustive and critical revision of the manuscript and for acknowledging the difficulties of the work. These comments moved us to significantly modify, carefully revised, (and hopefully) to improve the MS.
Presentation and clarity. First, the organization of the paper was deeply revised to improve clarity in the goals and messages of the paper. We wished to describe the complex (syndemic) nature of the AMR problem, the different ways to approach the problem in light of the new challenges in the 21st century, and how information must be carefully elaborated and communicated to different stakeholders to achieve their different goals and activities. In our experience, many professionals of different disciplines are not aware of the different dimensions of the AMR problem.
To improve clarity, we eliminated redundancies, clarified gaps and challenges, and detailed some specific aspects. The R1 version of the paper was divided into four complementary sections which can also be read independently. The first one reviews the socioeconomic factors that have contributed to building the current Global Healthcare system (new section), the scientific framework in which AMR has traditionally been approached in such a system, and the novel scientific and organizational challenges of approaching AMR in the fourth globalization scenario. The second discusses the need to reframe public health and global health in this context. Reviewer#3 made many comments about the content of these sections, especially what was sections 4 and 5 which are now combined in section 2.
Given that health and AMR policies and guidelines are supported by information from surveillance systems, in the third section we reviewed the unit of analysis (“the what” and “the who”) and the indicators (the “operational units of surveillance”) used in AMR and discuss the factors that affect the validity, reliability, and comparability of the information to be applied in various healthcare (primary, secondary, tertiary), demographic, and economic contexts (local, regional, global, and inter-sectorial levels). Finally, we discuss the disparities and similarities between distinct stakeholders’ objectives and the gaps and challenges of combatting AMR at various levels. In these sections, we tried to explain the gaps and the challenges to efficiently face the AMR problem. Comments by the referee regarding surveillance systems were addressed in these sections. The abstract, two sections, 1 table, and one figure are completely new.
Second, the manuscript was revised by a professional English proofreader.
The targeted audience. We adapted the text to scientific-minded lay-audiences using the framework described by Grundmann (section 4). Although the referee raised some criticisms about metrics, information, at any level, is essential for decision-making. We dedicated a section (section 3) to analyze the validity, reliability, and comparability of the results.
Specific comments and remarks
Part 1. and 1.1. This provides a historic account of the emergence and perception of AMR as an epidemic public health problem. The chapter ends with the notion that, thanks to advances in molecular epidemiology, the scope of the problem has been widened to include multiple bacterial species but also subgenomic genetic information in the form of mobile genetic elements (MGEs), without drawing conclusions on novel evidence-based interventions.
Author´s response. Thank you for the comment. As mentioned, the paper has been deeply revised. Nonetheless, your observation about the application of molecular epidemiology to novel interventions is wise and appropriate. We tried to address it in section 3 (challenges in the measurement tools), and section 4 (table 1-gaps and challenges and the corresponding text-section 4.1).
Page 2, line 1, title: consider changing “beyond” for “after”
Author´s response. The manuscript has been revised by professional English proofreader
“The Great Acceleration Period” One might guess that this encompasses the period of improving institutional health care from the likes of Nightingale/ Lister /Ogston until the great postwar bonanza of pills and profits when Pharmaceutical Industry-driven research started to market the majority of active compounds (which are still in use today) in the late 1950’s. Maybe this should be explained in the Glossary.
Author´s response. The Great Acceleration starts in 1950s. It defines the sharp increase of the human activities – population, economy, resource use, technologies – that drove rapid and unprecedented changes to the structure and functioning of the Earth System, the last one, the climate change (see also the recent UNEP document Bracing for Superbugs: Strengthening environmental action in the One Health response to antimicrobial resistance released by the UNEP last Feb8th (https://www.unep.org/resources/superbugs/environmental-action) . The ‘Great Acceleration becames the basis for a proposed new geologic epoch in Earth history, the Anthropocene (Steffen, W., Broadgate, W., Deutsch, L., Gaffney, O. & Ludwig, C. The trajectory of the Anthropocene: the Great Acceleration. Anthr. Rev 2, 81–98 (2015). The Great acceleration and the Anthropocene have been linked to the problem of antibiotic resistance in different papers (Living with Resistance Project. Antibiotic and pesticide susceptibility and the Anthropocene operating space. Nature Sustainability, 2018, vol. 1, no 11, p. 632-641; Gillings).
A novel section to explain the history of the healthcare sector within the socioeconomic context of the 20th century has been added (novel section 1.1). This section divides 20-21st centuries according the globalization waves. They explain the drivers of AMR beyond the great acceleration period. The term “great acceleration period” is now mentioned in the context of globalization and included in the “glossary table” following referee´s suggestion.
It would be useful for the reader to widen the Glossary at the end of the text and to include key words and concepts such as “death sinks”, “hothouses”, “consumptogenic systems”, “patches”, “hybrid lineages”, “orthogonality”, “abiotic reservoirs”, “market failure”, “evolution on a leash theory”. In addition, it would be helpful to add a parenthesis (see glossary) after the terms that have been included for explanatory purposes.
Author´s response. Done
page 2, line 5: “healthy environments” considering the large-scale outbreaks of infective jaundice (serum hepatitis) in the 1950’s in the US this may be a bit euphemistic.
Author´s response. The reviewer is right. However, we used the classical metaphors in seminal Dubos´ papers to highlight the enthusiasm by the implementation of antibiotics in the therapeutic arsenal. The text has been revised to put this expression in context.
page 2, line 20: replace “creating” by “helping”
Author´s response. The manuscript has carefully been revised so many expressions have been changed or eliminated.
Part 1.2. widens the view beyond health care institutions to include living conditions outside hospitals in an increasingly interconnected world. What I am missing here are a couple of examples that would enlighten the reader towards a better understanding of the arguments put forward. I also miss examples that show the “… abruptly and synergistically changed AMR bacterial species … “ (lines 24-25). The text still focuses (lines 30-32) on AMR in health care centers, although there are ample examples of community pathogens with increasing and relevant AMR phenotypes such as M. tuberculosis, Shigella spp. and S. typhi.
Author´s response. Thanks for the comment. Section 1.2 (now section 1.3.) has been carefully revised to avoid overestimations of some statements. The examples requested have been included. Some were mentioned in another part of the text and have now been moved here.
The chapter also gives little credit to the extent to which health care seeking behavior and individual or collective agency are driving forces of current trends. It demands the need for knowledge about disease ecology which should inform “… appropriate measures to combat / control infectious diseases and AMR…” but there are no real-world examples given.
Author´s response. The reviewer is right. This point is included in other part of the text but it is also mentioned here as a major challenge.
Page 3, line 49: Consider revising “This Review discuss how AMR heterogeneity is measure…”
Author´s response. The sentence has been completed. Also, a new section entitled “Challenges in Measuring AMR in the heterogeneous Healthcare network” (section 3) has been included. This part describes the gaps and challenges in the indicators and units of analysis we normally used and the issues regarding the validity, reliability, and comparability of the information. In the original version of the manuscript, we only discussed the use of some indicators and units of analysis. Although referee #3 raised some criticisms about the interest in the “metrics” for a general audience, we do believe that different types and levels of stakeholders face problems to interpret the available information provided by surveillance systems and policymakers. In section 3, we tried to explain the challenges in the measurement instruments in a general context. This section is complementary to others but can also be read independently.
Part 2. and 2.1. describes efforts of Public Health Agencies to focus attention and control efforts to predominant target organisms using so-called “watch lists”. It introduces the important observation that the burden caused by AMR in hospitals is “Pareto distributed” or when using mathematical terminology obeys power law dynamics. This pervasive symmetry cannot be emphasized enough, as it applies at every levels of genetic abundance i.e. Species, Clones and MGEs and is the crucial ingredient of all hierarchically distributed self-organizing networks.
Author´s response. Thank you very much for this comment. We have carefully revised the text taking it into account. This point is discussed in section 3 (validity and reliability issues).
It argues (lines 44-45) “The pandemic concept implies the widespread of the clones in the community” Apart from questionable syntax; I believe that this assertion is frankly wrong. Large-scale sampling and sequencing efforts across hospital and community boundaries have repeatedly demonstrated oligoclonality among the AMR isolates and high diversity among non-AMR carriage isolates.
This entire chapter would clearly benefit from a through revision especially linguistically. Some sentences are far too long (lines 4-13) and the semantics are easily lost to the reader. Again, the authors make the point that targeted interventions, given the complexity of AMR behavior are difficult to conceive but do not suggest implementations of any kind as promised in the title.
Author´s response. We agree with the reviewer's comments (“in absolute numbers”) which is aligned with Baas Bucking´ principles. Most of the members of a given taxa are minority populations but a very few are amplified and evolved after transmission and passing different host and stressors (in our case, those COPs lineages). However, the emergence and high occurrence of certain lineages may represent part of the basal microbiomes and under selection, may transmit to the close contacts (Faith JJ, Colombel JF, Gordon JI. Identifying strains that contribute to complex diseases through the study of microbial inheritance. Proc Natl Acad Sci U S A. 2015 Jan 20;112(3):633-40. doi: 10.1073/pnas.1418781112. Epub 2015 Jan 9. PMID: 25576328; PMCID: PMC4311841). The text has been modified for clarity and proofread by an English professional proofreader. We do believe that the concerns of the referee have been addressed.
Page 4, line 47: “hybrid lineages” are the result of large chromosomal replacements – maybe this should be mentioned?
Author´s response. The term has been included in the glossary as suggested in a previous comment.
Page 4, lines 49 -51: Sentence needs to be explained.
Author´s response. Done. Note that the R1 version is greatly modified and some parts could have been eliminated.
Part 2.2. purports to describe the heterogeneities among host populations who harbor AMR-pathogens or are more prone to infections and repeats to a certain extent what has been already mentioned under 1.2. Interestingly, it is argued, that the ecological disease understanding would invalidate Koch’s postulates in the light of the role of microbiota/microbiomes that may be conducive to disease expression. However, the same would hold for individual genetic host determinants as well as to immunity, both not mentioned in this paragraph. In a formal epidemiological sense, one would argue that alterations of the microbiome, hereditary traits as well as immunity might act as confounding variables on the expression of an infectious disease causally associated with specific pathogens.
Author´s response. Thanks very much for the comment. The text has been modified to include this idea. This part was moved to another section of the manuscript (new section 1.2, old section 1.1).
Page 5, line 28: Explain why “convenience” – would social network analysis be the key word here?
Author´s response. What we wished to highlight is the interest in elucidating the different levels of transmission paralleling the social levels used in other disciplines. This sentence has been deleted in the current version of the manuscript.
Page 5, lines 31 – 34: Sentence should be shortened. Why “AMR in Western health care centers” would that not apply to low resource settings as well?
Author´s response. We mentioned that because most information came from Western countries. We have eliminated the distinction between Western and other countries according to the referee´s comment.
Page 5, line 42: Superspreading is by itself equally “Pareto distributed”, so you may want to leave out “while other not”
Author´s response. The reviewer is right. The sentence has been revised according this comment.
Page 5, lines 45 – 52: The sentences pick up concepts of potential interventions but not in a constructive manner (how could an efficient public health response look like?). The second sentence is rather deconstructive, portraying the dangers of broad-brush interventions in LMICs. Explain: GBS = Group B streptococci?
Author´s response. The reviewer is right. We have revised the text according to the referee´s suggestion. We tried to be more constructive. On the other hand, it is difficult to suggest an alternative to MDAs in LMICs because we lack information on the effects due to the scarcity of monitoring analysis. Some gaps and challenges in this regard are included in the new Table 1. The GBS acronym has been changed to the name of the species.
Page 6, lines 1 – 4: revise end of the sentence
Author´s response. See previous comment.
Page 6, lines 5 – 8: This is rather a truism. Again, no mention of how this knowledge could lead to effective public health interventions.
Author´s response. What we wished to highlight was the need to analyze the AMR in homogeneous groups of patients to improve diagnostic and thus, the information of surveillance systems. This point is demanded by different stakeholders to define efficient antibiotic stewardship and infection control measures (doi:10.1093/JAC/DKAA426; doi:10.1093/JAC/DKAA428; doi:10.1093/JAC/DKAA427). The paragraph has been revised to avoid truisms. It is discussed in the context of the healthcare systems.
Part 2.3. elucidates the spatial heterogeneities and describes the process of dispersal from sources inside health care networks. This largely repeats notions already expressed in chapter 2.2.
Page 6, lines 13 – 14: “… following not only stochastic, but also deterministic processes… “ I don’t understand this. I thought that a deterministic process is the idealization (i.e. median behavior) of naturally occurring stochasticity?
Author´s response. The reviewer is right. We have deleted this sentence in the revised version of the MS.
Page 6, lines 14 – 20: I wonder, if this could be described in a bit more lucid manner?
Author´s response. As mentioned, the revised version of the MS has greatly been modified and the resulting version has been corrected by the English proofreader. The reviewer is right. We have deleted this sentence in the revised version of the MS.
The text also introduces the limitations of extant surveillance systems that focus on certain types of hospitals and indicator pathogens and describes the difficulties to extrapolate these findings for community action.
Page 6, lines 25 – 31: I find it difficult to believe that surveillance efforts that aim to quantify health determinants such as AMR, are designed to “tackle” disease. Foremost, they represent an early step towards measuring the size of a public health problem.
Author´s response. The reviewer is right. We have deleted this sentence in the revised version of the MS.
Page 6, lines 34 – 36: There are of course structural barriers to measure reality. For example, how would one go about measuring AMR in a Mumbai slum or in rural Tanzania?
I believe the biggest bias arises from the selection of patients for diagnosis. Diagnostic habits for many reasons are difficult to factor-in into the estimates. But this is more a technical issue and not caused by governmental interference: Governments before Covid found it very difficult to grasp the concept of measuring disease, at all. For vertical vs horizontal health interventions – see also the WHO Alma Ata Declaration of 1978.
Author´s response. Thank you very much for the comment. We have modified the text according to it. The declarations of Alma-Ata (and Astana) have been included in the text.
Part 3. reiterates the difficulties when quantifying AMR as a public health problem.
Page 6, line 51 – page 7, line 2: I would like to see the assertion that “… surveillance has played a pivotal role in early alerting of microbial threats, identifying transmission within and between health care centers, and monitoring and guiding the impact of different interventions… “ referenced. My experience is that surveillance systems are notoriously too sluggish and cannot be utilized for early warning and response, neither can the remaining claims be supported by experience and published sources. Microbial threats (such as novel ARGs) have usually been identified in clinical settings, when treatment failed and good diagnostic laboratories where at hand. Transmissions are usually ascertained during outbreak investigations or purpose-designed structured surveys, and success of interventions can hardly be discerned at the high aggregations level of the data that are used for surveillance purposes. There may be exceptions such as interrupted time series analyses, but usually these are deployed during intervention (quasi-experimental) trials and are not part of larger surveillance activities.
Author´s response. Thank you very much for the comment. The reviewer is right. We tried to modify the text according to it.
Page 7, line 3 – 7: “Gaps and challenges….” This sentence contradicts the first sentence “… requires a cost-effective and targeted sampling strategy.”
Author´s response. The reviewer is right. We have deleted this sentence in the revised version of the MS.
Part 3.1. The term Operational Unit of Surveillance (OUS) also within the three mentioned contexts (AMR, AMU and HAI) was first introduced by [PMID: 24694024; Ups J Med Sci. 2014 May;119(2):87-95. doi: 10.3109/03009734.2014.904458. Epub 2014 Apr 3.] and this should be referenced.
Author´s response. Thanks for this comment. The reference has been added. The careful lecture on this smart opinion article work moved us to include a new section to explain the gaps and challenges of AMR using the framework suggested by Grundmann in that paper ([PMID: 24694024).
Part 3.1.1. describes the tenet of international surveillance initiatives such as ECDC’s EARS-Net, ReLAVRA from PAHO/WHO and the WHO GLASS initiative that determines AMR prevalence on the basis of indicator drug-bug combinations in isolates from bloodstream infections and discusses the threats to validity of data generated in this fashion.
Page 7, lines 19 – 21: Sentence required clarification.
This chapter raises the issue of valid metrics (clinical breakpoints vs ECOFFs) for determining the prevalence of AMR as well the issue for the “correct?” endpoints. Although the text mentions proportion resistance among indicator organisms and clinical endpoints such as cause-specific morbidity and mortality as potential metrics, it however shies away from including a satisfactory discussion about the merits and complexities when choosing different endpoints for the sake of valid and comparable data generation by surveillance systems.
Author´s response. We have deleted this sentence in the revised version of the MS to avoid misinterpretations.
Page 7, line 30: Heavy metals are at present not used, neither for therapeutics (except for Leishmaniasis, and Trypanosomiasis) nor in infection control interventions.
Author´s response. The reviewer is right. The sentence has been deleted in the revised version of the MS.
The remaining text about AMR surveillance discusses the merits of including invasive isolates vs isolates from other sources. Rightly, the text argues that sampling (diagnostic habits) largely bias results and threatens the comparability across institutions as well as countries. Here the textbook example for the “death sin” in epidemiology is differential misclassification, when blood culture diagnostics is carried out as a last resort (when all treatment options have failed) and predictably, the surviving bacteria are likely to be of a MDR phenotype. Other sources of isolates such as screening isolates, environmental isolates as well as respiratory, intestinal and entire microbiomes are, to my knowledge, not included in surveillance systems. There are good reasons for the exclusion, but the text leaves a thoughtful discussion about the feasibility of these approaches to be desired for.
Author´s response. The reviewer is right. We have modified this part in the revised version of the MS to avoid confusions. It also appears in section 3 to address the problem of the reliability of the information.
The next section mentions the difficulties when measuring AMC/AMU with respect to stewardship interventions, especially in LMICs where AMU is confounded by the marketing of counterfeit and substandard compounds. This is altogether an important issue but requires a more detailed consideration which is beyond the remit of the current text. This should probably be mentioned to the reader.
Author´s response. The reviewer is right. The text has been modified according to his/her comment.
The same applies to the non-exhaustive nature of the discussion about surveillance of hospital-/health care-acquired infections (HAI). Here the text does not give sufficient room to the complexities of HAI surveillance.
This issue merits more detail and a more thoughtful discussion about the limitations, validity of endpoints as well confounding (such as underlying disease of patients treated in hospitals, which is not mentioned at all).
Mentioning of local ecological and socioeconomic factors is simply not sufficient to describe differential health care seeking behavior, financial imbursement systems, institutional quality management and governmental incentive systems, regional hospital network structures and hospital referral practices - all of which are impacting on the prevalence, diversity and composition of AMR pathogens in hospitals. Moreover, the biggest problem of HAI surveillance is how to define a nosocomial infection (including the arbitrary 48 hrs. rule, line 31), and the way in which case definitions and their interpretations undermine valid estimates of infection rates.
Author´s response. Thanks for the comment. We tried to address the reviewer´s concerns about the complexity of HAI surveillance in modern healthcare centers (see new sections 3 and 4) although we cannot go into detail due to the many different levels of the global healthcare network this review addresses.
Part 3.2. The argument that surveillance of environmental hotspots is not harmonized misses the point. There are innumerable technical and operational issues that need to be overcome and, to my knowledge, such surveillance systems do not exist. Finally, the most important question would be what would be the benefit of a surveillance of environmental hotspots for those that need to know? Are other risks in the environment, chemical, radiation, etc. not more pertinent issues to map? More important are initiatives that aim at community pathogens especially in LMICs and the text is right to mention this dearth but should include Shigella spp. and S. typhi.
Author´s response. The reviewer is right. The text has been modified according to his/her comment.
Part 3.3. Design of surveillance systems (including time-sensitivity of data) depends on stakeholder demands. What is needed by those who need to know? And what interventions could be informed by what kind of data? Are these time sensitive? A one-for-all surveillance system should be discouraged. I am missing a thorough discussion about the aims and objectives of AMR surveillance systems [see also PMID: 24694024; Ups J Med Sci. 2014 May;119(2):87-95. doi: 10.3109/03009734.2014.904458. Epub 2014 Apr 3.].
Author´s response. Thanks very much for the comment and suggestion. One of the major points of this article was exactly that “a one-for-all surveillance system” is not useful. We discussed how AMR guidelines and policies are based on the information provided by the surveillance systems, and how important is to get valid, reliable, and comparable information. This is now discussed in section 3. Furthermore, we included another new section #4 to highlight how different policymakers have different goals and require different information. This last section was inspired by the article PMID: 24694024 suggested by reviewer#3. The reading of that article was enlightening and moved us to include a new section to explain the gaps and challenges of AMR using the framework defined in that publication.
Part 3.4. This is more of the same and in the context of what has been discussed before a rather redundant enumeration of items. What I am missing are constructive suggestions for the utilization of surveillance data towards informed interventions.
Page 10, line 32, Title: Improve language. How does availability relate to AMR heterogeneities?
Page 10, line 43 – 44: What does triangulation mean in this context?
Page 10, line 49: Consider replaceing “scope” with rate.
Page 11, line 1 – 3: Sentences need to be revised.
Author´s response. Done. Note that the manuscript has been deeply modified and some sentences could be deleted. Also, the revised version corrected by a professional English proofreader.
Part 4. This chapter remains rather vague about the actual forces that shape the interactions that are describes in a very verbious manner.
Author´s response. This section has been removed.
Page 11, line 7 – 8. Consider rephrasing the first sentence.
Page 11, lines 17 – 19: Please explain what you mean.
Page 11, line 25: What precisely are ecosystem dimensions?
Author´s response. As mentioned, section 4 has been removed and thus these sentences were deleted too.
Part 5. and 5.1. Again, the text remains rather circumstantial without providing real world examples. It is stated, that interfaces are difficult to establish between human, animal and related environments, but the authors miss to mention studies that have tried to exactly do that – establishing the link between these OneHealth segments (using whole genome sequencing) and have quantified the net flux of AMR between these compartments. The authors also fail to mention that these studies (more often than not) were unsuccessful to establish genomic exchange across these interfaces. And therefore, the important scepsis about OnHealth syndemic scenarios is missing, rather the text recommends that “A multidisciplinary approach for the analysis of microbial metacommunities in a host metasystem landscape taking into account disparities across countries” needs to be established. There is no vision or idea mentioned on how this should or could be put into practice.
Author´s response. Section 5 has been moved and appears after section 1 to reflect the challenges of the 21st century (described in section 1) in Public Health and Global Health. The text has been modified and some sentences have been removed. We kept a paragraph to include the economic sectors in the discussion as it also appears in the last UNEP document and others. I hope to have removed vague or speculative parts and improved the text for clarity.
Page 13, lines 1 – 2: Revise sentence.
Author´s response. Done. The manuscript has been deeply revised and the current version corrected by a professional English proofreader.
Page 13, lines 11 – 14: Revise sentence. Explain how “market failure” adds to the problem.
Author´s response. We applied the expression “market failure” used in economics as a metaphor to highlight the negative effects of some interventions in some targeted sectors. We have eliminated the sentence (“The result is a current global economy based on systems of commercial success (wealthy systems) but market failure (negative effects on human and environmental health) that increasingly difficult decision making)” to avoid confusion.
Part 5.2. Much of what is written, becomes already clear from the previous text. The dichotomy of vertical public health approaches vs. horizontal “health promotion” was subject of the Alma Ata Declaration of WHO in 1978 and probably should be mentioned in this context. Some sentences lack logic.
Author´s response. Thank you very much for this comment. The Alma-Ata declaration has been included in the new section 1.1, and other parts in the text. Again, the manuscript has been deeply revised and the current version corrected by a professional English proofreader.
Page 13, lines 25 – 27: Clarify the semantics of this sentence, please.
Author´s response. Again, the manuscript has been deeply revised and the current version corrected by a professional English proofreader.
Page 13, 38 – 42: Improve the definition of the property of emergence “… as a phenomenon brought about by properties of a system, but cannot per se be predicted from the elements the system is made of…”
Author´s response. Done
Part 6. Conclusions
Revise ordinal number in title. Author´s response. Done
Page 14, line7: Change “on” for over. Author´s response. Done.

Round 2
Reviewer 3 Report
This is a comprehensive, albeit not yet exhaustive review about the dynamic emergence of AMR and its ramifications for surveillance and response. It nicely illustrates the authors’ encyclopaedic perspective about current and future challenges with regards to the control of AMR at different geographic and temporal levels. This viewpoint will certainly inform the current and pertinent scientific discourse. I laude the effort and have no hesitation to recommend publishing this improved version of the manuscript.
Author Response
Author´s Response General comment
We thank the authors for a careful and comprehensive revision considering all the aspects mentioned by the 3 reviewers. The article was majorly revised and restructured and as such all three reviewers and myself have given the revised manuscript a deeper look. I would like to address the following points for clarification and improvement. I appreciate that the article is an opinion paper and that some of the described aspects are personal opinions, so please consider most of my comments as an advice for the better understanding of the general reader rather than as a need or requirement for acceptance of the article. Here are the comments as follows:
Author´s response. We are extremely grateful for the comments made by the reviewers (especially the third one) and the editor which made us to revise and (hopefully) improve the article.
Individual comments
The reason to write some words and phrases in blue and in italics is obviously that they are fixed terms that are explained in the BOX 1. You should introduce this to the reader when it appears first, in line 48 (1st sentence) by writing “Antimicrobial resistance (AMR, see BOX 1 for definitions) is recognized …” . However, a number of these terms derive from areas far away from biology, medicine and public health and I cannot oversee if they are really set and fixed there. In addition, the reader might be confused instead of stimulated by these unusual terms or by terms that have a different interpretation in a medical or public health context (which is not meant here). I work myself for more than 25 years in the respected field and have never heard of terms like “individual agency”, “hothouses”, “patches”, etc. I am a bit uncertain if all these phrases are really fixed terms in their respective areas of origin and if at all, they are helpful in the context of this article. So my suggestion is to reduce the number of these terms to a list of phrases absolutely necessary for this article.
Author´s response. We have included a call to BOX 1 for all the terms which are further explained to make easier the reading to a non-medical audience. Note that one of the purposes is to cross terms and concepts between different AMR stakeholders. Thus, we would believe that introducing terms widely used in ecology, social sciences, and economy, is relevant for the purpose of this article. Nonetheless, we acknowledge the effort of the editor to make the article understandable to medical audience.
Line 133: I am not sure if it is the disease that evolves, but the pathogen. So maybe better “contributes to evolution of infectious agents (pathogens).” instead of “contributes to evolution of infectious diseases.”?
Author´s response. The text has been corrected accordingly.
Line 151: A “hospital outcome” could be manifold (e.g., profit) and for me it is not clear what the authors really mean here. Maybe better “…principles to improve hospital hygiene (performance?, safety?) in the United Kingdom…” instead of “…principles to improve hospital outcomes in the United Kingdom…”?
Author´s response. The text has been corrected accordingly.
Line 176: Did you mean “feedback loops “ instead of “feebback loops”?
Author´s response. It is a mistake. Nonetheless, the sentence has been modified to improve clarity. It now reads “..and also revealed the AMR transmission pathways between and within the One Health sectors with impact in the way to face the control and prevention of AMR” instead of “ and also revealed the AMR transmission pathways in the One Health sectors and the novel healthcare feedback loops with impact in the way to face the control and prevention of AMR”.
Line 179: “Afterwards,…” (s is missing)
Author´s response. Thanks. Corrected.
Line 181-186: The term “ecological disease” is not clear to me. Also, the term “Koch’s postulates” (in fact: “Henle-Koch’s postulates”) has not been introduced before and as such, what is meant with the “revision of Koch’s postulates” could only be clear to the informed reader. I am also challenging this aspect in this more general term, since not all aspects of Koch’s postulates are wrong in the context of HAI. For instance, the infectious agent can and should always be isolated from the infectious sample/source (postulate 1), it could be grown in vitro (postulate 2), but it may be complicated to induce an infection in a “susceptible microorganism” (postulate 3). Maybe it would be good to give this more space and discuss this aspect in more details in a separate paragraph (or to leave it out completely).
Author´s response. The reviewer is right. The text has been revised to address editor´s concerns. First, Koch’s postulates” has been changed by “Henle-Koch’s postulates”. Second, BOX 1 further explain Koch postulates for lay public.
Line 206: I challenge the limitation to “ in low incidence countries”. First, what you maybe mean is “spread(transmission) of … into low incidence countries”. Second, they may spread to the same extent “into high incidence countries”, but this is unrecognized due to its high general incidence. Consequently, I would skip the limitation to “in low incidence countries” or re-write it as “easily recognized when introduced into low incidence countries”. You could also mention wars resulting in patients’ transfers and massive waves of refugees as causes for transmitting MDR pathogens across borders and world regions (PMID: 22273504; PMID: 36695468; PMID: 36695452; PMID: 36695467).
Author´s response. We have revised the text to address editor´s requests. All the references suggested were included.
Lines 213-216: According to my textbook knowledge, terms like “epidemic”, “endemic” and “pandemic” are linked to infectious diseases explicitly. In the context of “non-communicable diseases” the term “pandemic” is misleading and wrong. This entire sentence is a bit confusing and its general message is unclear to me. What should be meant with “colossal global syndemic landscape”?
Author´s response. I understand this is a controversial issue. However, the link between NCDs and pandemics is increasingly used (especially for still births, Parkinson) and a high number of articles support that statement. (Allen L. Are we facing a noncommunicable disease pandemic? J Epidemiol Glob Health. 2017 Mar;7(1):5-9. doi: 10.1016/j.jegh.2016.11.001). We have included the term “pandemics” in Glossary-BOX 1 to explain and support with scientific references why is applied to NCDs.
Lines 217-219: This sentence is not intelligible either; please re-write to be clearer.
Author´s response. La sentence has been deleted.
Lines 226-236: The authors use phrases and words with an undefined meaning and since some of them are explained in BOX 1 they should be written in blue and in italics. The term “metacommunities” is neither explained in BOX 1 nor in Figure 1; please explain what it should mean (in a medical or scientific context). What should “bacterial migration” mean, in which context bacteria migrate? What should “microbiota coalescences” represent? I recommend to better define these evolutionary or philosophical terms and re-write the entire paragraph to be clearer.
Author´s response. The term metacommunities is increasingly used in the area of microbiome. Bacterial migration and coalescence are terms imported from (landscape) ecology. Since they are used once, we have deleted from the text to avoid an excess of terms that can result unfamiliar to general medical or public health audience.
Lines 242-243: I don’t believe that the “design of … efficient measures … is a challenge”. Efficient measures to combat AMR are well-described and recognized. It is always the “implementation” of these known measures under the given circumstances and conditions that challenge a successful application. Maybe you should re-write this sentence accordingly.
Author´s response. Done.
Line 246: maybe better “applied” or “implemented” instead of “used”?
Author´s response. Done
Lines 252-253: I think that several aspects are mixed up in this sentence. There is a reason to “diagnose a HAI” and this is to direct and initiate an efficient therapy for the individual patient. HAI and AMR surveillance is a subsequent subject that builds on a summary of individual cases diagnosed. The link between “HAI surveillance” and to “validate endpoints” (of what?) does not become clear here; please re-write and divide into several sentences if necessary.
Author´s response. Thanks for the comment. The reviewer is right. The sentence has been modified according to this comment.
Lines 255-256: I guess there are impressive and numerous “lessons learnt from the COVID pandemic”, but the consideration that there could be “more than one infectious disease at a time” (or what do you mean by “concurrent infectious disease”) is not one of the main challenges I would think about in this context. Please explain in more details what you especially mean here. Maybe you mean the link between the AMR crisis and the COVID pandemic which have worsened each other.
Author´s response. The sentence has been modified to address the editor´s suggestions.
Line 264: Why there are 2 references in brackets within the brackets for references?
Author´s response. It is a typo mistake. Corrected.
Lines 264-266: Maybe better “A scientific approach to explore and understand influences on behaviours … ” instead of “A behavioural science approach to explore and understand influences on behaviours … ”?
Author´s response. The sentence has been modified according to the editor´s comment.
Line 276: “hothouses” is a fixed term and should be written in blue and in italics(?).
Author´s response. The reviewer is right. The term is now defined in BOX1.
Line 279: Don’t know why “abiotic reservoirs” are risk factors for “AMR transmission”, what do you mean by this (hospital environment)?
Author´s response. The sentence has been modified to address the editor´s question. The term “abiotic reservoirs” has been changed to “environmental reservoirs of MDR pathogens”. This risk is now further explained in section 3.2 (Microbiomes and resistomes) and elsewhere (regarding attribution bias including Table 1).
Lines 322-323: I don’t agree with this statement. (AMR) surveillance systems provide data to benchmark a hospital, region or country regarding health-care infections (HAI) and AMR rates and frequencies with neighboring and comparable areas, but the “actions and policies” are not based on these data but on evidence-based studies and experiences gained demonstrating a successful reduction of the burden of AMR and HAI.
Author´s response. The first sentence of the paragraph has been deleted. Some lines below, we mentioned that the information provided by surveillance systems facilitates the design and implementation of measures to reduce the burden of AMR.
Lines 370: “The data collected ... are applied as basic … evidence about …” I don’t understand this assumption, either I have an evidence or not, but I cannot apply something as evidence(?!). Please rephrase.
Author´s response. The sentence has been deleted.
Lines 369: “indicator” is a fixed term in epidemiology and surveillance and does not require any further introduction of another definition/explanation. In addition, the term “Operational Unit of Surveillance (OUS)” which is introduced to explain “indicator” is not common, confuses the reader and is also not introduced in the cited reference [90]. Please delete and change in the text where it appears and in the tables.
Author´s response. We disagree with the editor´s comment. There are different types of indicators widely used in health (Input, output, impact indicators,see WHO definitions). Grundmann introduced the term “operational unit of surveillance” to further specify which indicators are used in surveillance of AMR. This “OUSs” term is widely used among the clinical microbiologist and authorities in the field.
Finally, we disagree with the editor about the suitability of the reference 102 to support the term “OUS“. The concept is introduced in such reference 102 (see pag 90, first line “scope section): “Operational unit of surveillance (OUS) is the marker or determinant for a health state which is recorded and reported”. We revise the text to improve clarity of these terms (indicators, unit of analysis,…).Two footnotes are also added (12 and 13, page 8).
Line 417: I guess that “(ref)” suggests that there is a reference missing?!
Author´s response. A reference has been included.
Pages 10/12: the line break appears at a wrong place within a sentence and leaving half a page blank. Please modify.
Author´s response. Done.
Lines 448-453: This is a very long sentence and I guess that there is a verb missing(?). Should “difficult” (line 452; an adjective) be substituted by “complicate” (a verb)?
Author´s response. The sentence has been revised according the editor´s suggestion.
Line 457: Not sure if the term “intestinal and respiratory microbiome” is a good choice in this sense. I would prefer and suggest “intestinal, oral and skin microbiome” or better use “mucosal” instead of “respiratory”.
Author´s response. We disagree with this comment. The respiratory tract microbiome (RTM) constitutes a continuous ecosystem with a longitudinal and transversal gradient of microbial diversity from the nasal and oral cavities to the alveoli. This is what also defines the gastrointestinal microbiome (GIM). RTM and GIM differ in biomass, diversity and taxonomic composition (Bassis C. M., et al.. (2015). Analysis of the upper respiratory tract microbiotas as the source of the lung and gastric microbiotas in healthy individuals. MBio 6, 1–10. 10.1128/mBio.00037-15; Pérez-Cobas et al. Altered Ecology of the Respiratory Tract Microbiome and Nosocomial Pneumonia. Front Microbiol. 2022 doi: 10.3389/fmicb.2021.709421; Pérez-Cobas et al. Ecology of the respiratory tract microbiome, Trends in Microbiology, in press).
Lines 459-460: The “intervention” is the “fecal transplantation” (instead of “transplant”).
Author´s response. The editor is right. Corrected.
Lines 467-468: AMC and AMU also describe “how antimicrobials are dispensed and prescribed in the community setting” (not only “healthcare facilities”). In many parts of the world, antimicrobials are sold without any prescription (over the counter) and control, so the community setting cannot be neglected in this discussion.
Author´s response. The editor is right. The sentence has been modified to address this aspect and supported by references.
Lines 487: Please explain “bystander selection” in one sentence. It is not intelligible what you mean by this phrase/sentence by simply linking the given references?
Author´s response. The sentence has been modified to address this comment.
Lines 497-505. This is a complicated paragraph. Fixed terms that are not self-explanatory are explained by additional fixed terms that come along with a special meaning and that are again not self-explanatory. The less informed and qualified reader does not understand what the authors really mean. Please try in simpler wording. A general comment: AMR surveillance by classical methods (AST) and simple approaches (only BSI) has been successfully implemented on a regional, country or international level (EARS-Net, GLASS, etc). This is without any doubt a reliable and substantial source of information relevant to identify AMR trends and to justify and proof the success of interventions. The limitations could be mentioned, but it is relevant to address not only the latter without specifying or mentioning the successes and benefits of a system which has its limits. On the contrary, by knowing the limits, it is possible to better judge and evaluate the quality of available data and to clarify for what they are useful and for what not.
Author´s response. We feel sorry the message was not understood. This section was not focused on what is already used but on the pitfalls that influence the validity of such information. We agree that BSI has been widely used in different surveillance programs but, as mentioned in footnote 22 this samples cannot be used to measure AMR in healthcare institutions of primary and secondary levels where the occurrence of severe infections is really low.
Lines 521-523: I don’t understand the meaning of this sentence. How can a “cause” be equivalent to the term “data attribution”? Please modify and rewrite.
Author´s response. We used “cause” as the equivalent of “class” because we thought would be clearer to the reader. The word “class” is defined as “a set or category of things having some property or attribute in common and differentiated from others by kind, type, or quality”. The assignment of an observation to a particular class is not always obvious. Sometimes, observations are attributed to classes that are known to be wrong (because the most adequate class is unknown). In other cases, observations can be attributed to multiple classes at the same time. The data attribution bias is really frequent in AMR surveillance because we used the same indicators for a disparate number of “units of analysis” and do not have into account its “validity”. Some examples are given in footnote.
In summary, the word “cause” has been changed by “class”. The sentences have been slightly modified to improve clarity. References support the statement.
Pages 14/16: the line break appears at a wrong place leaving almost the entire page blank. Please modify.
Author´s response. Thanks. Done.
Line 555: I cannot find a place where HIC is introduced as an abbreviation; please explain here.
Author´s response. Thanks. Done.
Line 591: Please divide “OneHealth”.
Author´s response. Done.
I am not sure what I should understand under the phrase “to facilitate the evolution of AMR.”? It sounds like that one could direct evolution using settings that “facilitate” or “complicate evolution of AMR”. What means “evolution of AMR” at all, emergence of AMR in a previously susceptible pathogen or population, increase of MICs, accumulation of AMR determinants? Please specify.
Author´s response. We believed that this explanation is unnecessary due to the huge number of emblematic quotes that consider and explain in detail that 1) hospitals are hotspot for disease and AMR evolution, and 2) AMR is the most relevant example of evolution. We will only mention two very popular books among infectious diseases and clinical microbiologists
- About 1). In a 1958 paper entitled, ‘‘The Evolution of Infectious Diseases in History’’, and using a metaphor borrowed from epidemiologist William Farr, René Dubos compared infectious diseases to weeds and medicine to gardening. Doctors were’’ like gardeners whose work never ends, students of disease must always be on the lookout for new problems of infection’’ (Dubos 1958, p. 450). “For hospitals, this meant that while they had been intended as hygienic environments, mass antibiotic therapy had turned them into hothouses of disease evolution, where uncommon and dangerous infections thrived on a uniquely vulnerable population”. (Dubos, R. The evolution of infectious diseases in the course of history. Canadian Medical Association Journal, 1958; 79(6), 445–451).
- Regarding 2) Stuart Levy commented, ‘‘Antibiotic usage has stimulated evolutionary changes that are unparalleled in recorded biologic history’’ (Levy, S. B. (1992). The antibiotic paradox. How miracle drugs are destroying the miracle. New York,London: Plenum Press.). Others by Julian Davies mention similar points.
Line 599: Isn’t the 48 h time span to differentiate between an ambulant (<48 h) vs nosocomial acquisition (>48 h) of an infection a European definition? I thought that in Northern American studies the time window is </>72 h?
Author´s response. The editor is right. The sentence has been modified.
Line 612/613: I disagree. The colonization status of discharged patients with MDR pathogens into the community is not “unknown”. There are several studies following the discharge of patients into the community and the family/private household setting documenting short term and log term colonization with MDR pathogens over weeks and months. Admittedly, these studies are not numerous, small in size and coverage, but existing and far from being “unknown”.
Author´s response. Of course, the “colonization status” of discharged patients with MDR pathogens into the community has been analyzed in some papers. No doubt about it. As the editor highlights, the number of publications is scarce and we only wished to highlight that we cannot extrapolate fractionated evidences to a general context. The sentence has been modified to avoid misunderstood.
Lines 615/616: I think that the recognition of the AMR problem is nowadays very high on the political and strategic agendas of all big international players (WHO, UN, EU, ECDC, CDC, OIE, G7, G20, etc). So, I personally disagree with the first sentence. Maybe you meant to “contain the AMR problem” instead of “to place the AMR problem”; but then you should name it as such.
Author´s response. The editor is right. The sentence has been modified as the editor suggests.
Line 633: did you mean “extended life expectancy”?
Author´s response. We imagine that this is line 663 instead of 633. The editor is right. The sentence has been modified as the editor suggests.
Table 1. What do you mean by “pharmacovigilance sales sources”? Pharmacovigilance describes the post-licensing surveillance of the efficacy and safety of a pharmaceutical product (here potentially an antibiotic). In addition, one could count “sales” of an antibiotic, but this is not linked to the previous term and quantity is already addressed in the AMC/AMU terms and phrases. Please explain.
Author´s response. The editor is right. The text has been corrected.
Lines 681-683: Maybe change “decision-making and guidelines” to “recommendations and guidelines”?
Author´s response. Done.
I have commented on this point earlier (Lines 322-323). Guidelines and recommendations to prevent and control AMR in the healthcare sector build on expertise gained and studies performed with a clear focus on hygiene aspects (and not from data from clinical labs and surveillance systems). Corresponding recommendations receive a grading and come along with a value for each individual measure (value of (i) hand hygiene, (ii) single bed rooms, (iii) daily antiseptic bathing, (iv) wearing gloves, etc). Each of these measures is tested individually in studies and finally receives a strength of scientific reliability (what is the difference between wearing gloves vs not wearing gloves in HAI pathogen transmission between and nosocomial infections in patients, when otherwise hand hygiene and all other factors are identical/at the same level between study populations). I recommend changing this statement.
Author´s response. Thanks for the comment. The text has been corrected.
BOX 1: “Hybrid Lineages” appears 2x with two slightly different explanations.
Author´s response. The mistake has been corrected.
Hybrid lineages/”hopeful monsters”: A “hybrid lineage” is a new lineage of an (micro)organism genetically recombined out off two previous lineages. I guess this is understandable for the general reader. The term “hopeful monsters” was obviously introduced in the early 20th century and mentioned in the opinion paper cited from 1934 [118]. Goldsmith wrote that he understands it as a lineage (monster) “which would start a new evolutionary line if fitting into some empty environmental niche.” I personally could hardly set these two terms as equivalent to each other due to a number of reasons. First, Goldsmith is an evolutionary biologist and has a different view on the subject than a physician or a microbiologist. Second, the general knowledge and understanding of genetics in the early 20th and early 21st century is completely different (Goldsmith had no clue about “hybrid lineages”). Third, a new hybrid lineage of a HAI pathogen does not necessarily “fit into an empty niche”, but a niche that is already taken by other pathogens and commensals. Forth, a single acquisition of a small and simple marker like a resistance gene (cassette) or a virulence plasmid could build a new entity and a new “hopeful monster”, but I would not consider this as a “hybrid lineage”, but as a strain that has acquired a resistance cassette or a virulence plasmid. Please reconsider and/or comment.
Author´s response. We disagree with the editor on some points. The term “hopeful monsters” coined by Goldsmith has been used as a metaphor for the “saltational” form of evolution. Hybridization is one, but the acquisition of a trait that deeply changes the adaptation of a microorganism is another one. The acquisition of a gene conferring resistance to antibiotics is not a saltational form of evolution (at least, in a general way). Instead, the acquisition of citrate by Escherichia coli in Lenski experiments which enable the microorganisms to change the habitat, is.
The adoption of the Goldsmith terminology “hopeful monster” is nicely illustrated in Chouard, T. Evolution: Revenge of the hopeful monster. Nature 463, 864–867 (2010). It has been also used by Klugman and Croucher to illustrate the emergence of Klebsiella pneumoniae ST258 in an editorial of mBio (Croucher NJ, Klugman KP. The emergence of bacterial "hopeful monsters". mBio. 2014 Jul 29;5(4):e01550-14. doi: 10.1128/mBio.01550-14).
Despite our disagreement, we understand that this could be controversial and we do not need to add Goldsmith concepts to the paper. Thus, the term hopeful monsters have been eliminated.